# Unified Scaling Laws for Compressed Representations

**Andrei Panferov**[*]
ISTA

**Alexandra Volkova**[*]
ISTA

**Ionut-Vlad Modoranu**
ISTA

**Vage Egiazarian**
ISTA

**Mher Safaryan**
ISTA

**Dan Alistarh**[†]
ISTA & Red Hat AI

## Abstract

Scaling laws have shaped recent advances in machine learning by enabling predictable scaling of model performance based on model size, computation, and data volume. Concurrently, the rise in computational cost for AI has motivated model compression techniques, notably quantization and sparsification, which have emerged to mitigate the steep computational demands associated with large-scale training and inference. This paper investigates the interplay between scaling laws and compression formats, exploring whether a unified scaling framework can accurately predict model performance when training occurs over various compressed representations, such as sparse, scalar-quantized, sparse-quantized or even vector-quantized formats. Our key contributions include validating a general scaling law formulation and showing that it is applicable both individually but also composably across compression types. Based on this, our main finding is demonstrating both theoretically and empirically that there exists a simple "capacity" metric—based on the representation's ability to fit random Gaussian data—which can robustly predict parameter efficiency across multiple compressed representations. On the practical side, we extend our formulation to directly compare the accuracy potential of different compressed formats, and to derive better algorithms for training over sparse-quantized formats. Our source code is available at: `IST-DASLab/unified-sc-laws`

## 1 Introduction

A key recent advance in machine learning has been the idea of *predictable scaling* of learning performance with respect to *model, computation and data sizes*. This approach is encompassed by the *scaling laws* Kaplan *et al.* [17], which allow researchers to predict the values of these three parameters required to reach a certain model performance. This powerful idea has been expanded upon in several directions, e.g. [15; 5; 22], and is a key ingredient behind the massive expansion of computational power for AI [12].

A parallel research direction, motivated by this massive increase in computational cost, has been *model compression*, which proposes a series of techniques to reduce the computational and memory footprint of model inference and training, via techniques such as *sparsification* [14] and *quantization* [11]. In this paper, we focus on the interplay between scaling laws and the degree of compression of the representation over which learning occurs. While there is significant emerging work in this direction, e.g. [9; 19; 33; 31], current scaling laws are specialized to single representations (e.g., quantization or sparsity) and/or formats (e.g., integer quantization), and cannot yet address the question of predicting model scaling behavior when training over general compressed representations.

---

[*]Equal contribution.
[†]Correspondence to: dan.alistarh@ist.ac.at.

39th Conference on Neural Information Processing Systems (NeurIPS 2025).

**Contributions.** This paper is structured two main questions, and their practical ramifications:

**Q1: Is there a unified compression scaling law?** First, we wish to find a single general law that not only applies to sparse [9] or quantized [19] representations in isolation, but that also provides a good fit for *hybrid formats*, such as sparse-and-quantized weights, or *compound compression*, i.e. sparse weights *and* activations. Through extensive experimentation, we identify this law to be of the form

$$Loss(N, D) \sim A \cdot (N \cdot \rho(R))^{-\alpha} + B \cdot D^{-\beta} + E, \tag{1}$$

where $N$ is the number of model parameters, $D$ is the dataset size, $E$ is the irreducible error, $A$, $B$, $\alpha$ and $\beta$ are constants, and $\rho$ is a parametric function of the representation $R$. Crucially, we find that, even for very complex representations—e.g. 3-bit quantization with group size 32 and 1% outliers in full-precision—the parametric function $\rho$ can still predict the scaling of model performance w.r.t. the parameter count $N$. We call $\rho(R)$ **the representation capacity** of $R$. Consequently, there is always a "dense equivalent" parameter count $N' = N \cdot \rho(R)$ which would yield the same loss during training. The capacity $\rho(R)$ lies naturally in the interval $(0, 1]$, and the key goal of compression is to maximize the trade-off between model accuracy and the size and computational cost of the representation.

**Q2: Is capacity an "intrinsic" property of the representation?** While related forms of the above law have been proposed in prior work [10; 19], we are the first show that capacity is an intrinsic property of the representation, independent of the model and task for which the scaling law is obtained, but relatable to standard information-theoretic measures. Moreover, we establish the applicability of the law across hybrid (e.g. sparse-quantized weights) or composite (e.g. quantized weights-and-activations) representations.

More precisely, our main finding is that *capacity is tightly-correlated with the representation's ability to fit random Gaussian data, measured in terms of minimal mean-squared error (MSE)*. Concretely, $\rho(R)$ is a simple parametric function of the $MSE$ of the representation $R$ when fitting random Gaussian data, i.e. $\rho(R) = \tilde{\rho}(MSE(R))$, where instances of the same representation $R$, e.g. 3 and 4-bit integer quantization, *share the same parametric form $\tilde{\rho}$*. This finding, which we validate across quantized, sparse, quantized-sparse, and even vector-quantized representations, provides a simple metric to "rank" different formats implementing the same representation. In addition, this also allows us to determine the "optimal" capacity at a certain bit-width, which is given by theoretical bounds on Gaussian fitting for a given support, which can be easily estimated via Monte Carlo algorithms. In addition, we also provide a non-trivial theoretical justification for this relationship in Theorem 1, for Adam-optimized compressed models: we relate the convergence of Adam over compressed representations with the product between the number of parameters $N$ and the average root mean-squared error of compression across optimization, which connects to our notion of capacity.

Our second finding is that, except for pathological cases, *capacity factorizes across composite representations*: concretely, the capacity of a 4-bit and 2:4 sparse model is the product between the capacity of the 4-bit dense model, and that of a 2:4-sparse but unquantized model. Factorization allows us to evaluate the capacity of complex representations based on simple ones, and also holds when compressing different model representations, e.g. both weights and activations.

**Practical Implications.** The analytical metrics suggested by representation capacity also have non-trivial practical applications. First, the fact that we are able to relate the predictive parameter $\rho$ to intrinsic properties of the underlying representation gives us the ability to *analytically predict the representational power of different compressed numerical formats*. This way, we can accurately compare and predict the efficacy of various formats such as Floating-Point, Integer (INT with and without grouping), or sparse-quantized formats (2:4 + INT) at different compression budgets. Second, this framework inspires an improved approach for sparse training, which we show provides significant improvements (above 20% in some sparsity regimes) in capacity at the same number of parameters.

Overall, our results provide a new lens to view the scaling properties of compressed models, with respect to intrinsic properties of the representation over which training is performed. Thus, we believe that capacity-aware scaling has the potential to become a practical design principle for the next generation of efficient foundation models.

## 2 Preliminaries

**Scaling Laws.** We start from the "Chinchilla" scaling law formulation [15] that proposed to model loss scaling as a function of the number of parameters in the model $N$ and the number of data points

Table 1: Representation scaling laws (rows) versus the quantities of interest (columns). For all laws, $N$ represents the number of parameters, $D$ is the data, and $E$ is the irreducible error. For the sparsity scaling law of Frantar *et al.* [8], $S$ is the sparsity and the lowercase parameters are learnable constants. For the precision scaling law of Kumar *et al.* [19], $P_w$ is the weight precision, and $\gamma_P$ is a learnable weight sensitivity parameter. For the law of Frantar *et al.* [10], $\text{eff}_C$ is the "effective parameter multiplier," that is explicitly fitted for every instance of compression $C$. By contrast, our formulation postulates that the parameter efficiency is a simple parametric function of the representation's capacity to fit random Gaussian data ($GMSE(R)$).

| Parametrization | Formulation for $Loss(N, D)$ | Sparsity fit (Error) | Quantization fit (Error) |
|---|---|---|---|
| **Sparsity** $S$ 
 Frantar *et al.* [9] | $\dfrac{a_S(1-S)^{b_S} + c_S}{N^{b_N}} + \left(\dfrac{a_D}{D}\right)^{b_D} + E$ | $5.7 \cdot 10^{-4}$ | N/A |
| **Quantization to** $P_w$ 
 Kumar *et al.* [19] | $A\left[N(1 - e^{-P_w/\gamma_w})\right]^{-\alpha} + BD^{-\beta} + E$ | N/A | $4.5 \cdot 10^{-3}$ |
| **Compression** $C$ 
 Frantar *et al.* [10] | $\dfrac{A}{(N \cdot \text{eff}_C)^\alpha} + \dfrac{B}{D^\beta} + E$ | $4.2 \cdot 10^{-4}$ | $1.9 \cdot 10^{-3}$ |
| **Representation** $R$ 
 (OURS) | $\dfrac{A}{(N \cdot \widetilde{\rho}(GMSE(R)))^\alpha} + \dfrac{B}{D^\beta} + E$ | $4.7 \cdot 10^{-4}$ | $2.1 \cdot 10^{-3}$ |

$D$ the model was trained on, in the form the parametric function:

$$Loss(N, D) = AN^{-\alpha} + BD^{-\beta} + E, \tag{2}$$

where $A, B, E, \alpha,$ and $\beta$ are the scaling law parameters that can be fit empirically. It is important to note that such scaling laws assume an ideal, well-tuned training setup, and that the parameter may vary slightly depending on architecture, optimizer, and hyper-parameters.

**Compressed Representations.** For *sparsity*, we assume that a specific fraction, within each parameter group of a certain size $G$, is set to zero. Sparsity is *unstructured* if the group is the whole tensor, whereas it is semi-structured (N:M) if $N$ parameters out of every $M$ are set to zero. For *quantization*, unless otherwise stated, we assume that parameters are mapped onto a *scalar, symmetric* grid corresponding to the number of bits available for quantization, as is standard [11]. (We will also consider vector quantization in Section 4.1.) For *sparse-quantized* representations, we follow [13] by first applying sparsification, and then quantization, to map continuous parameters onto this format.

**Prior Scaling Laws.** The relationship between the learning representation and the the scaling law formulation was considered by Frantar *et al.* [9] for sparsity, and by Kumar *et al.* [19] for quantization. The scaling laws they propose are described in Table 1, together with their parametrization, for the special case of weight-only compression. While both these laws can predict loss with respect to training over their target representations, their formulation is not designed to generalize to other representations, or to hybrid ones (e.g. sparse-quantized).

The unified law we consider extends preliminary work by Frantar *et al.* [10], who, assuming that training happens over weights compressed in representation $C$, proposed a simple parametric law similar to Equation 1, but which is fitted independently for each instance of compressed training, yielding a value of the corresponding parameter efficiency factor, called $\text{eff}_C$. Frantar *et al.* [10] focuses on quantization; they fit sparsity in limited experiments, and do not consider hybrid formats.

**Our Approach.** By contrast, our focus is on relating parameter efficiency to intrinsic properties of the representation $R$: in their parlance, we show that, across all instances of a given compressed representation $R$, e.g. uniform integer (INT) quantization, the parameter efficiency has the same parametric form $\rho(R)$, and, in fact, this parametric form is simply a function of the MSE for the representation $R$ w.r.t. random Gaussian data, i.e. $\rho(R) = \tilde{\rho}(GMSE(R))$. Importantly, $GMSE(R)$ is an intrinsic property of $R$, and only depends on its own parametrization: for INT, this would be the number of bits we employ per parameter.

The fact that this parametric form is shared across instances of the same representation (Section 4.1), is powerful since it allows us to compare and transfer parameters between instances of the same representation $R$. Clearly, if $GMSE \simeq 0$, then $\rho(P) \simeq 1$, and we recover the original "dense"

scaling law [15]. Interestingly, Table 1 shows that our unified law can provide a better fit than the representation-specific formulations of Frantar *et al.* [9] and Kumar *et al.* [19], and (almost) matches the formulation of [10], which is fitted for each compression instantiation $C$.

**Setting for Experimental Validation.** For our scaling law investigations, we pretrained decoder-only Transformers following the Llama architecture [34] for 30M, 50M, 100M and 200M non-embedding parameters. The models were trained on the C4 dataset [28], using the Llama-2 tokenizer [34]. To ensure we operate in a data-rich regime, we use 50, 100, and 200 training tokens per model parameter for each training configuration, and train on fixed-length context windows of 512 tokens. We used AdamW [18; 23] with a 0.1 ratio of warm-up epochs with cosine scheduler. Our experimental setup is very similar to that of [9; 19; 10]. More details are provided in Appendix A.

We follow standard quantization-aware training (QAT) methods, combined with various levels of unstructured weight sparsity. For quantization we employ the gradient estimator of [27], a per-layer uniform quantizer with static scaling factors and gradient masking. Quantization levels range from 1-bit to 8-bit precision. We consider configurations with quantized weights only, activations only, and both simultaneously. For sparsity, we apply unstructured magnitude pruning via top-k thresholding on a per-layer basis. The sparsity mask is recomputed dynamically at each optimization step. For Vector Quantization (VQ), we follow QuEST scalar quantization and apply it to 2- and 4-dimensional HIGGS grids [24]. To restrain outliers we use the trust estimation method [27] that zeros out gradients for any point lying outside a hypersphere of a certain radius.

## 3 Theoretical Analysis

One key focus of our work is whether, given a compressed representation $R$ over which learning is performed, we can identify a predictive metric that correlates with the representation's efficiency $\rho(R)$. To identify this metric, we first model the standard weight-compressed LLM optimization process, which combines the Adam algorithm [18] with the straight-through estimator (STE) [2]. We have:

$$
\begin{aligned}
\text{STE Gradient:} \quad g_t &= \widetilde{\nabla} f(\widehat{\theta}_t), \qquad \text{where } \widehat{\theta}_t = \mathcal{C}(\theta_t), \\
\text{Optimizer states:} \quad m_t &= \beta_1 m_{t-1} + (1 - \beta_1) g_t \\
v_t &= \beta_2 v_{t-1} + (1 - \beta_2) g_t^2, \qquad \widetilde{v}_t = \max(v_t, \widetilde{v}_{t-1}) \\
\text{Model update:} \quad \theta_{t+1} &= \theta_t - \eta \frac{m_t}{\sqrt{\widetilde{v}_t} + \epsilon},
\end{aligned}
$$

where $\widetilde{\nabla}$ represents stochastic mini-batch gradient operator, $\mathcal{C} \colon \mathbb{R}^N \to \mathbb{R}^N$ is the compression scheme, $\beta_1, \beta_2 \in (0, 1)$ are momentum parameters, $\epsilon > 0$ is a small constant for numerical stability and $\eta > 0$ is the learning rate or the step-size. All vector operations are element-wise, including the $\max$ operation. Our analysis relies on the following assumptions, which are standard in adaptive optimization [36; 29; 4; 20; 6; 25; 30]:

**Assumption 1** (Lower bound and smoothness). *The loss function $f \colon \mathbb{R}^N \to \mathbb{R}$ is lower bounded by some $f^* \in \mathbb{R}$ and is L-smooth, namely, $\|\nabla f(\theta) - \nabla f(\theta')\|_2 \le L \|\theta - \theta'\|_2$, for any $\theta, \theta' \in \mathbb{R}^N$.*

**Assumption 2** (Unbiased and bounded stochastic gradient). *For all iterates $t \ge 1$, the stochastic gradient $g_t$ at $\widehat{\theta}_t$ is unbiased, $\mathbb{E}[g_t] = \nabla f(\widehat{\theta}_t)$, and uniformly bounded, $\|g_t\|_\infty \le G_\infty$, by some constant $G_\infty \ge 0$.*

**Assumption 3** (Bounded variance). *For all iterates $t \ge 1$, the variance of the stochastic gradient $g_t$ at $\widehat{\theta}_t$ is uniformly bounded by some constant $\sigma^2 \ge 0$, namely $\mathbb{E}[\|g_t - \nabla f(\widehat{\theta}_t)\|_2^2] \le \sigma^2$.*

In this context, our main claim is the following:

**Theorem 1** (Non-convex convergence analysis). *Let Assumptions 1, 2 and 3 hold. Then, choosing step-size $\eta = \min(\eta_0, \frac{1}{\sqrt{T}})$ with $\eta_0 = \frac{\epsilon(1-\beta_1)}{2LC\sqrt{N}}$ and $C = 2\sqrt{G_\infty^2 + \epsilon/N}$, a randomly chosen compressed iterate $\widehat{\theta}$ from $\{\widehat{\theta}_1, \ldots, \widehat{\theta}_T\}$ satisfies*

$$
\mathbb{E}[\|\nabla f(\widehat{\theta})\|^2] \le \frac{CLG_\infty}{\sqrt{\epsilon}} \cdot \mathbb{E}\left[\frac{1}{T} \sum_{t=1}^{T} \|\widehat{\theta}_t - \theta_t\|_2\right] \cdot N + \frac{C\sqrt{N}}{\sqrt{T}}\left(f(\theta_1) - f^* + \frac{L\sigma^2}{\epsilon}\right) + \mathcal{O}\left(\frac{N^{3/2}}{T}\right).
$$

**Discussion.** Similar convergence analysis for Adam under compressed iterates $\widehat{\theta}_t$ was performed in the setup of convex online learning with bounded domain condition [16], and in nonconvex

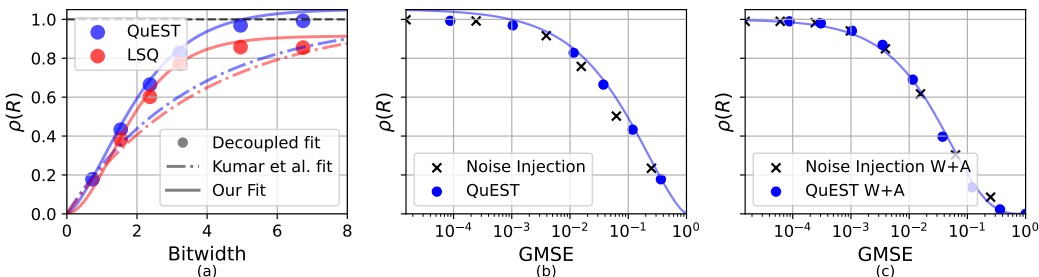

Figure 1: Comparison of $\rho$ fits for scaling law forms from Table 1: **(a, left)** shows quantizations scaling laws, **(b, middle)** and **(c, right)** demonstrate the match between noise injection and QuEST quantization for weight-only and weights+activations quantization.

optimization with restricted hyperparameter choices and slower rate in terms of constants and extra log-terms [3]. We now interpret this bound, whose complete proof can be found in the Supplementary. Specifically, the bound shows the ergodic convergence of the gradients taken at compressed iterates, which is the strongest property shown even in the uncompressed case [36]. In turn, this term is bounded by 3 terms on the RHS. The second and third bounding terms are standard in the analysis of uncompressed Adam [36], showing that our analysis is fairly tight. We focus our attention on the first term, whose key part is highlighted in blue: this term consists of absolute constants, multiplying the critical term $N \cdot \mathbb{E}\left[\frac{1}{T}\sum_{t=1}^{T}\|\widehat{\theta}_t - \theta_t\|_2\right]$. Specifically, this term consists of the number of parameters $N$ times *the average $\ell_2$ compression error over the model parameters throughout the execution*. Thus, this analysis suggests that the parameter efficiency of such models may depend multiplicatively on both the number of parameters, and the compression root-mean-square-error (RMSE) throughout the execution. Importantly, this bound is independent of the compression type.

## 4 Findings

### 4.1 Finding 1: Gaussian RMSE Predicts Representation Capacity

Table 1 presents a number of scaling laws that model the same functions via different parametrizations. One can notice, that both the *Sparsity* form of Frantar *et al.* [9] and the *Quantization* form of Kumar *et al.* [19] can be reduced to the *Decoupled* form of Frantar *et al.* [10] in the third row, by imposing additional constraints (e.g. $\text{eff}_P = 1 - e^{-P_w/\gamma_w}$ for quantization). Naturally, the Decoupled form can achieve lower fit error, but it does not provide any information about the interpretation of the capacity term, which we call $\rho(R)$, across different representations $R$. The *Sparsity* form and the *Quantization* form, on the other hand, feature intertwining and interpretable parameters. For simplicity, we first focus on the *Quantization* form for now.

**The Functional Form.** Kumar *et al.* [19] choose the functional form $\rho(P_w) = 1 - e^{-P_w/\gamma_w}$ to model quantization efficiency. By contrast, we propose a different form to model $\rho(R)$:

$$\widetilde{\rho}(GMSE(R)) = L \cdot \tanh(F \cdot \log_{1/4}(GMSE(R)))^C, \qquad (3)$$

which depends only on the representation $R$'s Gaussian-MSE fit, denoted by $GMSE(R)$, and on the scalars $L$, $F$, and $C$, detailed below. The $GMSE(R)$ is easily computable for any representation, and allows us to bypass the dependency on representation-specific parametrization, such as bit-width or sparsity. Specifically, we fit the scalar parameters for each compression type, e.g. scalar quantization, and then re-use these parameters while varying $GMSE(R)$ w.r.t. compression parameters, e.g. bit-width. The scalar parameters $L$, $F$, and $C$ allow us to accurately model observed effects such as:

- **Imperfect convergence in high-precision:** While modern QAT algorithms such as QuEST reach efficiency $\rho = 1$ for low quantization error, older algorithms such as LSQ (Figure 1 (a)), have an efficiency limit strictly below 1, since for instance its gradient estimator introduces consistent bias. The factor $L$, defaulting to 1 for saturating representations, allows us to model this imperfection.

- **Various low-precision curvature:** As seen in Figures 1 (b) and (c), different representations behave differently around $GMSE = 1$, with some have noticeably higher curvature ("breakdown").

From Figure 1 (a), one can see how that region disproportionally affects the law of Kumar *et al.* [19], leading to a very poor fit at higher bitwidths. The factor $C$, closer to 1 for representations "more linear" around $GMSE = 1$, allows us to more accurately model $\rho(R)$.

**Quality of Fit.** Table 1 shows that our approach leads similar or better quality-of-fit relative to prior laws, covering both scalar quantization and sparsity, while Figure 1 shows $\rho(R)$ alignment between scaling law forms, compared to Kumar *et al.* [19], for the QuEST and LSQ quantizers. Again, our approach provides significantly better fit. In Figure 2(a), we show that our method can also provide a good fit for models trained with vector-quantized (VQ) weights, using the projection method of [24], for lattice dimensions 2 and 4. This shows both the versatility of our approach, and the necessity of the $L$ term, since higher-dimensional VQ appears to have clear sub-unit saturation due to higher bias. We provide further examples in Section 4.3.

## 4.2 Finding 2: Noise Injection as a Scaling Law Predictor

Next, we turn our scaling law on its head, and ask: what if we plug the *optimal* achievable $GMSE$ for a certain bit-width into the scaling law? In that case, the scaling law should allow us to compute a *lower bound* on the achievable parameter efficiency given a certain type of representation. In turn, we can find out how close existing quantization- or sparsity-aware training techniques, or numerical formats, are to the information-theoretic lower bound for that specific representation.

Figure 1 (b) illustrates the "optimality gap" for the QuEST algorithm for scalar weight-only quantization across bit-widths, suggesting that this approach is fairly close to optimal. In Figure 1 (c), we compare the fit between actual runs of this QAT algorithm across bit-widths, and the predicted values via noise injection [1] (plugging in the equivalent $GMSE$) into the scaling law, showing a near-perfect fit.

## 4.3 Finding 3: Representation Capacity Is Multiplicative Across Compression Types

In prior work, [19] have claimed that, for their formulation of the law, the representation capacity factorizes independently for quantization of weights and activations. Our experimental findings extend this result, showing that representation capacity, $\rho(R)$, also factorizes naturally across a wide range of compression approaches, whether for the same tensor (sparse-and-quantized weights) or for different state tensors (sparse weights and sparse activations). We follow the experimental setup from Section 2, training models with sparse weights and activations, or sparse-quantized weights, or both. We fit a scaling law in the 100 toks/param regime. We show that representation capacity factorizes for the following scenarios:

1. **Sparse weights and activations:** For sparsity, independently applied to weight and activations,

$$\rho(R_{s_w, s_a}) = \rho(R_{s_w}) \cdot \rho(R_{s_a}). \tag{4}$$

   We summarize the fitted values of $\rho(R)$ levels in a matrix $M$ (Figure 2(b)), where each entry corresponds to the fitted efficiency for a model trained with a specific sparsity configuration. Remarkably, the matrix can be accurately approximated by a rank-1 outer product of the first column $M_{0,:}$ (weight-only) and the top row $M:, 0$ (activations-only) elements, i.e. $M \approx M_{0,:} \otimes M_{0,:}$. The resulting parameter efficiencies closely match the product of efficiencies obtained for runs with weight-only and activations-only configurations.

2. **Sparse and quantized weights:** Given a weight sparsity level $s$ combined with $q$-bit quantization, we claim that the representation capacity can be represented as the product: $\rho(R_{q,s}) = \rho(R_q) \cdot \rho(R_s)$. We report the results for different sparsity levels and bit width in Figure 9. Similarly, the matrix $\rho(R)$ factorizes into the outer product of marginal vectors for quantization-only and sparsity-only representation. Apart from extreme quantization to 2-bit precision, the approximation maintains the error of order of $10^{-2}$.

3. **Sparse and quantized weights, and quantized activations:** Finally, we observe that factorization extends to quantization of activations as well. In supplementary experiments, we apply quantization to activation tensors alongside with weight sparsity and quantization. Our results indicate that the representation capacity with weight sparsity $s_w$ and quantization bitwidth $q_w$, and activation sparsity $q_a$ follows $\rho(R_{q_w, s_w, q_a}) = \rho(R_{q_w}) \cdot \rho(R_{s_w}) \cdot \rho(R_{q_a})$.

This result allows for low-cost comparison across compression comparison. Moreover, it facilitates compression hyperparameter tuning and thus predictable model training in a compressed regime.

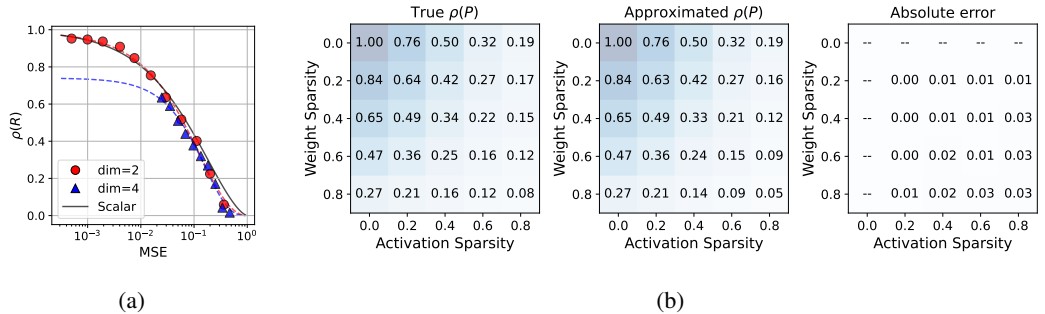

|  | (a) | | (b) |

Figure 2: **(a)** Scaling law for 2- and 4-dimensional vector quantization. **(b)** Representation capacity across weight and activation sparsity levels: baseline, factorized prediction, and relative errors. Note the low errors for the factorized predictions, with slight increases at the larger sparsity levels.

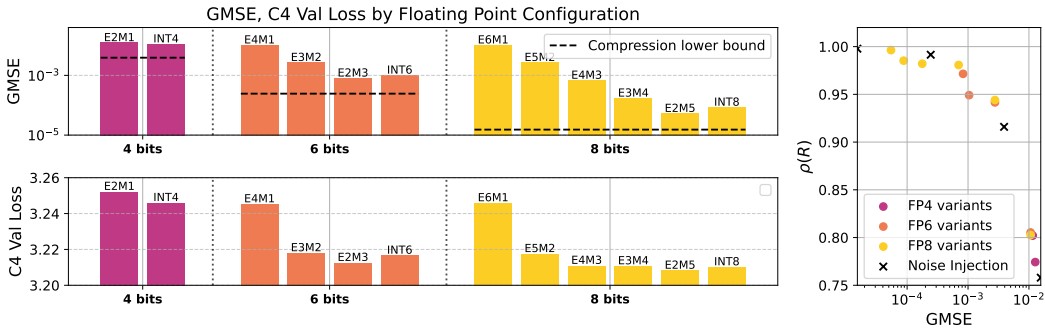

Figure 3: Comparison of floating point and integer data-types in terms of $GMSE$, and C4 Validation Loss when trained using the corresponding formats via QuEST, and the resulting capacity $\rho(R)$. Observe the high correlation between ranking in terms of $GMSE$ (top), and Val. Loss (bottom).

## 5 Applications

### 5.1 Application 1: Comparing Compressed Numerical Formats

**Practical Formats.** The scaling law for compressed representations enables systematic comparison of numerical formats used in quantization, such as INT8, INT4, FP4, or custom low-precision representations, based just on their $GMSE$, which can be determined via fast Monte Carlo algorithms. Thus, it provides a clear guidance on which low-precision format delivers the best performance for given resource constraints. Figure 3 illustrates this for a number of floating-point and integer

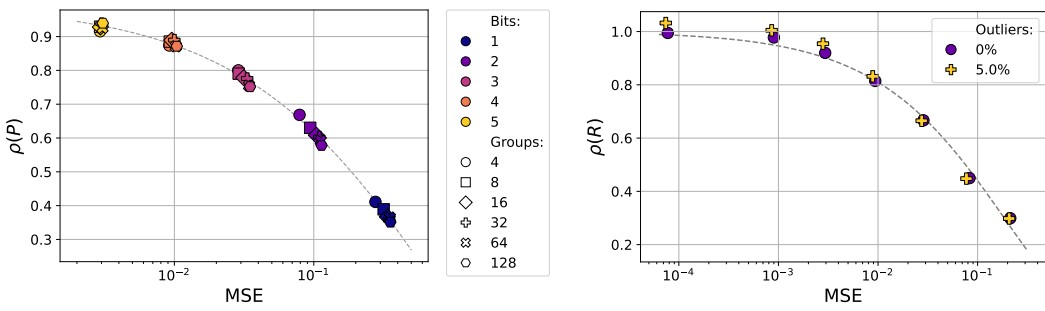

Figure 4: Representation capacity $\rho(R)$ versus MSE for **(a)** group-wise quantization, with markers indicate group counts (color encodes quantization bitwidth), and **(b)** outlier-aware quantization.

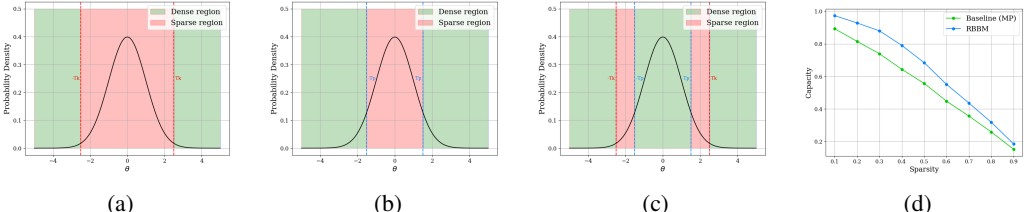

(a)          (b)          (c)          (d)

Figure 5: Backward heuristics: (a) forward mask $M_{FW}$ determined by the $T_k$ threshold, (b) backward mask determined by threshold $T_p$, (c) banded backward mask determined by both $T_k$ and $T_p$, (d) capacity plot compared to the Magnitude Pruning baseline with $M_{BW} = M_{FW}$.

data-types. Specifically, we observe a direct correlation between the ranking of $GMSE$ values (top) and the actual C4 validation loss obtained in actual experiments (bottom). This suggests that our $GMSE$ metric is an accurate predictor of compressed pre-training performance. For instance, it suggests that switching to FP4 (E2M1) will not bring gains relative to INT4 training, and that both formats are close to the theoretical lower bound at 4 bits.

**The Impact of Parameter Grouping and Outlier Preservation.** A related question regarding formats is whether more complex approaches, such as group-wise quantization, or outlier preservation in higher precision, can disrupt the scaling law. We examine this in Figure 4, which it shows that preserving no outliers (0 %) lies on the Pareto-optimal boundary: higher outlier ratios achieve a worse trade-off between the MSE and the representation capacity $\rho(R)$. This suggests that, for pre-training it is more effective to allocate bits to encoding the values distribution rather than outlier preservation or careful grouping. This further demonstrates that the proposed RMSE dependency is a general property and remains valid even under diverse structured compression techniques.

**Compositionality.** An immediate practical application of the multiplicative behavior of the law (Section 4.3) is the ability to estimate the model's performance in advance for arbitrary compression configuration. Given the individual efficiencies of different compression methods, such as quantization or sparsity, applied to weights or activations, one can predict the combined effect without spending additional compute for training.

## 5.2 Application 2: Increasing the Capacity of Sparse Training

**The Sparse Training Problem.** For our second application, we investigate implications of the RMSE law to maximize the capacity of a sparse representation during training. Specifically, standard sparse training methods such as gradual magnitude pruning (GMP) [37; 9] compute a forward sparsity mask, which we denote by $M_{FW}$ during the forward pass based on the absolute-magnitude Top-K operation applied to the model parameters $\theta$ with respect to a target sparsity. Then, a gradient $\nabla L(TopK(\theta))$ is taken w.r.t. the sparsified weights. Standard baselines, such as the ones we use for sparse training, re-use the forward sparsity mask for the backward, preventing the pruned weights from receiving any gradient. We are interested in heuristics to improve the parameter efficiency of this standard approach, increasing capacity at the same sparsity level.

**RMSE-Banded Gradient Masking.** For this, we follow the RMSE law and align the parameters $\theta \in \mathbb{R}^N$ with the standard normal distribution by dividing $\theta$ by its root mean square $RMS(\theta) = \sqrt{\frac{1}{N}\sum_{i=1}^{N}\theta_i^2}$, which results in $||\theta/RMS(\theta)||_2^2 = N$. We allow the user to provide a median deviation parameter $p \in (0, 0.5)$, which determines the threshold for the backward mask $T_p = RMS(\theta) \cdot ppf(0.5 + p)$, where $ppf$ is the inverse cumulative distribution function of the standard Normal distribution. The multiplication by $RMS(x)$ "converts" the threshold for the standard Normal distribution to the threshold for the vector $\theta$. As a result, $M_{BW} = |\theta| > T_p$.

Effectively, our approach, which we call RMSE-Banded Backward Masking (RBBM), sets a backward mask $M_{BW}$ that may be different than the TopK mask for the forward, whose sparsity is controllable via the parameter $p$. To address the fact that it may not allow gradient flow for small parameter values, we allow gradients to flow for the smallest and largest parameters, and create a band between $T_p$ and the TopK threshold $T_k$, where we do not allow gradient flow. Let $m = min(T_p, T_k)$ and $M = max(T_p, T_k)$ and define $M_{BW} = (|\theta| < m) \lor (|\theta| > M)$. Since we do not control the

relationship between $T_p$ and $T_k$, we need to ensure that the band is defined correctly. Concretely, the values $\theta_i < m$ and values $\theta_i > M$ will receive gradients, while the values $\theta_i \in [m, M]$ will not.

To illustrate, in Figure 5a we show the structure of forward mask $M_{FW}$, were the red region corresponds to values $|\theta_i| < T_k$ that will not receive any gradient, while the green region corresponds to the values $|\theta_i| \geq T_k$ which will receive gradient. The top-k threshold $T_k$ is fixed. In Figure 5b, we have a similar behavior for the backward mask $M_{BW}$ determined by the threshold $T_p$, which is now user-controlled via the median deviation parameter $p$. In Figure 5c we show an example for $T_p < T_k$, where we obtain a banded-mask: the values $|\theta_i| \in [T_p, T_k]$ will not receive any gradient (red region), while the other values will receive gradient (green region). The band width can be controlled via the parameter $p \in (0, 0.5)$. When $p$ is close to zero, the $T_p$ value will decrease, having the effect of increasing the width of the red band, where the corresponding weights do not get gradient. When $p = 0$, the value of $T_p$ will be equal to the median and this will be equivalent to the baseline (e.g. $M_{BW} = M_{FW}$) illustrated in Figure 5a.

**Results.** We apply RBBM for sparse training in our pretraining scenario, for the 30M parameter Llama model, using our training setup from Sec. 2, and for unstructured sparsities between 10% and 90%. We compute the capacity of the sparse representation. The results for our RMSE-based heuristic and the standard sparse training baseline (Magnitude Pruning) are provided in Figure 5d. The results show that our RMSE-based approach enables consistently higher capacity than the baseline.

# 6    Related Work

We focus on studies that extended classical scaling laws [17; 15] to model compression. Frantar *et al.* [9] presented the first scaling law for *weight-sparse Transformers*, across vision and language and unstructured and semi-structured sparsity. Earlier work by Clark *et al.* [5] studied mixture-of-experts sparsity, deriving scaling laws in terms of total parameters and compute per token, reinforcing the idea that only effective parameters govern scaling.

A recent breakthrough by Kumar *et al.* [19] introduced scaling laws that incorporate numerical quantization during training and inference, showing that, as for sparsity, a model trained in low precision behaves like a smaller high-precision model. They also apply their approach to post-training quantization (PTQ), showing that PTQ quality *worsens* as training data increases. For training-time quantization, their laws suggest that using lower precision allows training larger models on the same compute budget. Relative to their pioneering work, we bring the following improvements. First, we investigate a different and arguably simpler scaling law, showing that it yields a considerably better fit for quantization itself (see Table 1). Second, our key focus is different, we provide a first interpretation of the notion of representation "capacity", together with a theoretical justification, and ample experimental validation. Finally, we validate the factorization property posited by Kumar *et al.* [19], as well as extensions to hybrid formats. Follow-up work by Sun *et al.* [33] examines scaling laws for floating-point (FP) formats, finding that the law of Kumar *et al.* [19] does not provide a good fit in this case, and investigates an extension of the law via additional parametrization.

Preliminary work by Frantar *et al.* [10] proposed the single-parameter scaling law on which we build, and showed that it can be applied to instances of weight quantization and sparsity, by directly fitting the efficiency parameter. By contrast, we identify a *general* law, in the sense that the same parametric form can transfer between compression types, to hybrid sparse-quantized formats, as well as to instances where both weights and activations are compressed. More interestingly, we equate the representation capacity factor in the law with a natural notion of representation capacity, show that the law factorizes across representations. In concurrent work, ParetoQ [22] aimed to unify the fragmented landscape of LLM quantization by systematically evaluating the interplay between training strategy, quantization function design, and bit selection. Our results complement their findings: for instance, we obtain that, for the architectures we consider, 2-bit weight-only quantization is Pareto optimal.

# 7    Discussion and Limitations

Our study introduces *representation capacity*—roughly defined as a simple monotone transform of the Gaussian MSE—as a unified metric when training compressed models across various representations. Capacity enables format comparisons without retraining or exhaustive grid searches, so that future hardware designers can expose any format whose capacity $\rho$ dominates the Pareto frontier, confident

that software will exploit it optimally. Moreover, our law *factorizes*, further simplifying the search for the "optimal" training format.

A few caveats remain. First, in line with prior work in this area, our experiments are limited to decoder-only Llama-style architectures trained on C4 in the data-rich regime (100 toks/param); we plan to extend this at larger scale. Second, the law may need specific fits for ultra-low precision (e.g. 2-bit or ternary formats) and for vector-quantization codebooks below 8 entries, suggesting second-order effects may need to be taken into account. Third, while our theoretical evidence uses standard assumptions, it could be extended to more complex representation types.

## Acknowledgments

We would like to thank Lambda Cloud for their generous computational grant. We thank the NVIDIA and Google corporation for their grants, which supported part of this research. Alexandra Volkova and Andrei Panferov were supported in part by the BILAI Cluster of Excellence program.

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

## Appendix Roadmap

This appendix provides supporting material organized as follows:

- **Experimental Setup** (Appendix A): Model architectures, hyperparameters, and training configurations.
- **Factorization of Representation Capacity** (Appendix B): Detailed analysis showing how representation capacity matrices can be factorized for various compression techniques including quantization, sparsity, and their combinations.
- **Ablation Studies on Law Formulation** (Appendix C): Investigation of different noise distributions (Gaussian, Logistic, Student's t, Laplace) and functional forms (tanh, logistic) for the scaling law formulation.
- **Scaling Laws for Vector Quantization** (Appendix D): Implementation details and algorithms for vector quantization approaches, including forward and backward pass descriptions for HIGGS-based training.
- **Theoretical Support** (Appendix E): Convergence analysis for Adam optimizer with Straight-Through Estimation (STE), including complete proofs and supporting lemmas.
- **Improved Sparse Training via RBBM** (Appendix F): Comparison of our backward mask heuristics against RigL and Gradual Magnitude Pruning, with detailed descriptions of different masking strategies.

## A    Experimental setup

**Hyperparameters.** Table 2 summarizes the architectural and optimization hyperparameters used in this study.

| Model size | # Layers | # Heads | # Embeddings | Learning rate |
|---|---|---|---|---|
| 30 M | 6 | 5 | 640 | $1.2 \cdot 10^{-3}$ |
| 50 M | 7 | 6 | 768 | $1.2 \cdot 10^{-3}$ |
| 100 M | 8 | 8 | 1024 | $6 \cdot 10^{-4}$ |
| 200 M | 10 | 10 | 1280 | $3 \cdot 10^{-4}$ |

Table 2: Key architectural and training hyperparameters for Llama family models.

**Generalization Across Model Families and Datasets.** To validate that our findings are independent of the model family and dataset, we conducted comparable experiments on the OLMo2 [26] family models trained on the ClimbMix dataset [7]. OLMo2 models use no biases, employ rotary positional embeddings [32], RMSNorm [35], and reordered pre-normalization [21; 26]. We also used ReLU$^2$ activation function for linear layers.

We matched the model size range, data-model size ratios, and optimizer hyperparameters to those used in our original setup. Scaling laws of the same functional form as in Table 1, where $\rho(\text{GMSE})$ takes the form of Eq. 3, were refitted from scratch. We report the corresponding estimates for parameters $(\alpha, \beta, E, L, F, C)$ in Table 3. Confidence intervals of one standard deviation were obtained by the bootstrapping procedure. The results match closely with those from the Llama / C4 setup, with overlapping confidence intervals.

We use 8x80GB H100 machines for efficient training, and training one model takes on average 1 hour. To produce the full set of results we ran in total approximately 250 such training runs for various compression configurations.

## B    Factorization of Representation Capacity

Figures 6-9 show factorization of the representation capacity matrix for various in-training compression techniques:

1. Quantized weights and activations (Fig. 6).

|       | Llama family / C4 | OLMo2 family / ClimbMix |
|-------|-------------------|-------------------------|
| $\alpha$ | $0.13 \pm 0.05$ | $0.18 \pm 0.03$ |
| $\beta$  | $0.33 \pm 0.06$ | $0.26 \pm 0.02$ |
| E     | $1.3 \pm 0.5$   | $1.4 \pm 0.3$   |
| L     | $1.0 \pm 0.7$   | $0.84 \pm 0.10$ |
| F     | $0.41 \pm 0.02$ | $0.37 \pm 0.04$ |
| C     | $1.39 \pm 0.08$ | $1.24 \pm 0.13$ |

Table 3: Scaling Law hyperparameters for Llama and OMLo2 model families.

2. Sparsity + QuEST quantizer (Fig. 7).

3. Joint sparse & quantized weights + activations (Fig. 8), for all combinations $(s_a, q_a, q_b)$ for sparsity $s_a \in [0.25, 0.5, 0.75]$ and bit widths $q_a, q_b \in [2, 4, 6]$.

4. Sparsity + uniform quantizer with maximum absolute value as a scale (Fig. 9).

From the factorized representation-capacity matrices we observe the following:

1. The element-wise error of the fitted coefficients $\rho$ (from our scaling law) is of order $10^{-3}10^{-2}$.

2. A rank-1 row-column outer product accurately approximates the matrix, confirming the multiplicative property of representation capacity $\rho$ in various scenarios.

3. Approximation error remains of the order $10^{-2}$, except for the cases of *extreme* 2-bit quantization, where $\rho \lesssim 0.1$. We explain this gap due to the poorer performance of the optimizer in these extreme compression regimes, which is not taken into account currently by our model (as it uses the same coefficients for both 16 and 2 bits).

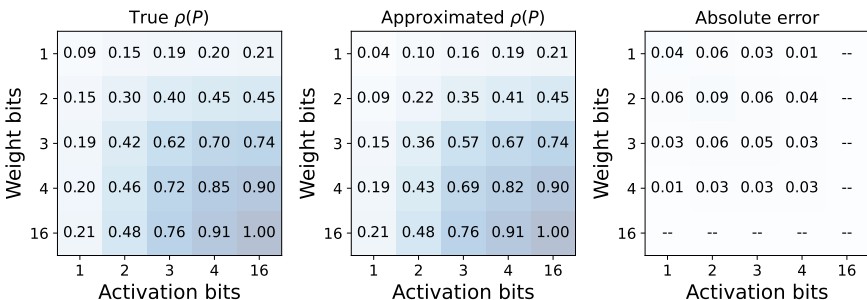

Figure 6: Representation capacity coefficients for independent quantization of weights and activations. Element-wise $\rho$ fitting error is not greater than $5 \cdot 10^{-3}$.

## C   Ablation studies on Law Formulation

### C.1   Evaluating RMSE across Different Distributions

We investigate how the choice of noise distribution used in our law formulation from Sec. 4.1 affects the predicted representation capacity. In Figure 10a we plot the mapping $\rho(MSE)$ for different bit widths using Logistic, Student's t, and Laplace noise distributions. Each distribution is rescaled to have zero mean and unit variance.

We observe that, no matter which noise distribution we choose, the mapping $\rho(MSE)$ always remains strictly monotonically decreasing. In principle, one could use heavytailed distributions (for example, Student-t or Laplace) to give more weight to extreme outlier errors. However, this leads to a smaller range of MSE values. By contrast, assuming Gaussian noise—which we propose—produces the widest spread of MSE, which in turn allows for a better fit for the scaling law. In short, although

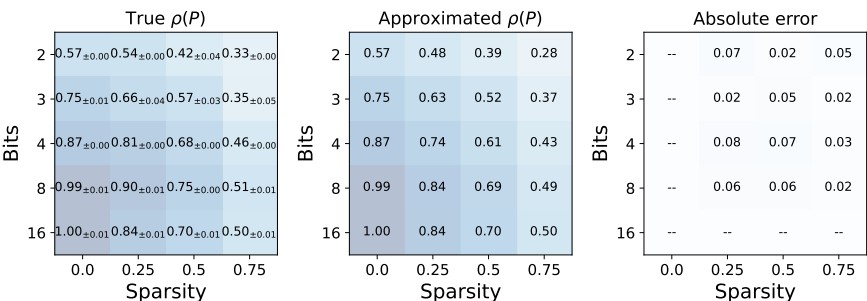

Figure 7: Representation capacity coefficients with fit errors in case of sparsity combined with the QuEST quantization.

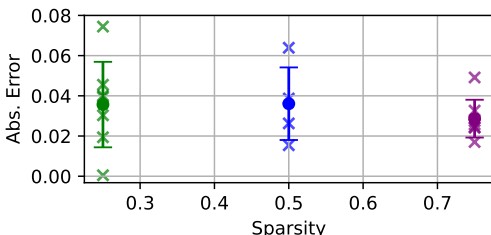

Figure 8: Representation capacity fit errors for sparse+quantized weights and quantized activations. Error bars denote $\pm 1$ standard deviation from the mean.

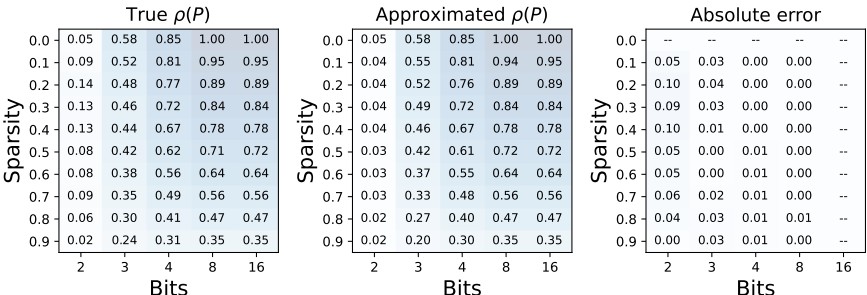

Figure 9: Representation capacity coefficients matrix for sparsity applied with uniform quantization. Element-wise $\rho$ fitting error is not greater than $2 \cdot 10^{-3}$.

monotonicity is preserved under various distributions, the Gaussian MSE delivers the best overall representation capacity prediction, so we adopt it as our default formulation.

Throughout this work, unless specified otherwise, MSE is computed over standard Gaussian input.

### C.2 Functional form of the Law

The behavior of $\rho(GMSE)$ observed in our experiments can be captured by fitting multiple smooth, monotonically decreasing functions, with no more than 3 additional parameters. In principle, a wide range of such functions can be used to model this relationship, depending on the desired fit properties.

For lower overall fitting error, we found it beneficial to constrain the function to satisfy boundary conditions $f(0) = 1$ and $f(1) = 0$. This way the correct behavior in the high-error region $MSE \lesssim 1$ is enforced, which is critical for stable predictions in the extreme compression cases. The corresponding fits are summarized in Table 4, the fitting error is calculated for the combined scaling law $L(MSE) = L(\rho(MSE))$.

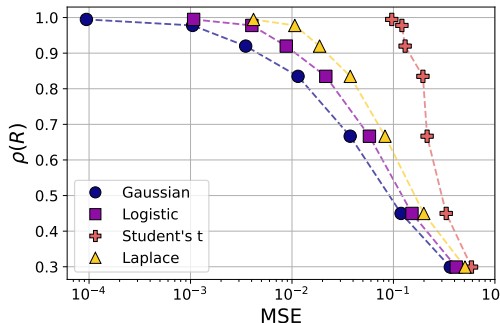 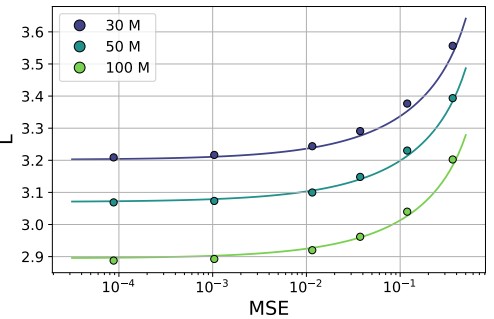

(a) Effect of input noise distribution on the mapping $\rho(MSE)$.

(b) Different functions used to fit $\rho(MSE)$

| | | Functional form | Fitting error (RMSE) |
|---|---|---|---|
| Decoupled | | Independently fitted $\rho$ | $4.2 \cdot 10^{-4}$ |
| Tanh | | $\rho = \tanh(F \cdot \log_{1/4} \mathrm{MSE})^C$ | $4.7 \cdot 10^{-4}$ |
| Logistic | | $\rho = (1 + B \cdot \mathrm{MSE}^A)^{-1}$ | $4.9 \cdot 10^{-4}$ |
| Logistic $(1, 0)$ | | $\rho = \dfrac{1 - \mathrm{MSE}^A}{1 + B \cdot \mathrm{MSE}^A}$ | $1.1 \cdot 10^{-3}$ |

Table 4: Functional form choices and associated fitting error.

Throughout this work, we adopt the functional form of hyperbolic tangent as it provides the smallest fitting error.

# D   Scaling Laws for Vector Quantization

In this section, we provide detailed information about the Vector Quantization approach used to produce the results in Figure 2(a). Algorithms 1 and 2 describe the forward and backward passes over a linear layer actively quantized with HIGGS for row-major weights. As was described earlier, our method is combines ideas from Panferov *et al.* [27] for the gradient estimator, and Malinovskii *et al.* [24] for the lattice representation. We use the trust estimation method that zeros out gradients for any point lying outside a hypersphere of radius $R$: $\|x\|_2^2 > R^2$. Our experiments were conducted on 30M and 50M models using the same set of hyperparameters as in Sec. 2.

---

**Algorithm 1** `VQ Training Forward`

---

1: **Input:** Input activations $\mathbf{x}$, row-major weight $\mathbf{w}$
2: $\mathbf{w}_h = \mathrm{HT}(\mathbf{w})$
3: $\hat{\mathbf{w}}_h = \mathrm{proj}_{grid} \mathbf{w}_h$
4: $\mathbf{y} = \mathbf{x}\hat{\mathbf{w}}_h^T$
5: **Return:** $\mathbf{y}, \mathbf{x}, \hat{\mathbf{w}}_h, M_{grid}(\mathbf{w}_h; \hat{\mathbf{w}}_h)$

---

# E   Theoretical Support

Here we provide the full proof of Theorem 1 giving a convergence analysis of the Adam optimizer when used with STE. For completeness, the description of the algorithm is presented in the Algorithm 3.

---

**Algorithm 2** `VQ Training Backward`

---

1: **Input:** $\frac{\partial L}{\partial \mathbf{y}}$, $\mathbf{x}$, $\hat{\mathbf{w}}_h$, $M_{grid}(\mathbf{w}_h; \hat{\mathbf{w}}_h)$
2: $\frac{\partial L}{\partial \mathbf{x}} = \frac{\partial L}{\partial \mathbf{y}} \hat{\mathbf{w}}_h$
3: $\frac{\partial L}{\partial \hat{\mathbf{w}}_h} = \mathbf{x}^T \frac{\partial L}{\partial \mathbf{y}}$
4: $\frac{\partial L}{\partial \mathbf{w}} = \text{IHT}\left(M_{grid}(\mathbf{w}_h; \hat{\mathbf{w}}_h) \odot \frac{\partial L}{\partial \hat{\mathbf{w}}_h}\right)$
5: **Return:** $\frac{\partial L}{\partial \mathbf{x}}$, $\frac{\partial L}{\partial \mathbf{w}}$

---

---

**Algorithm 3** Adam with Straight Through Estimation (STE) and AMSGrad normalization

---

1: Input: parameters $\beta_1$, $\beta_2 \in (0, 1)$, $\epsilon > 0$, step-size $\eta > 0$, $\theta_1 \in \mathbb{R}^d$, $m_0 = v_0 = \tilde{v}_0 = 0_N$
2: **for** $t = \{1, 2, ..., T\}$ **do**
3: $\quad \hat{\theta}_t = \mathcal{C}(\theta_t)$ $\qquad\qquad\qquad\qquad$ ◇ Compress the model via quantization and/or sparsification
4: $\quad g_t = \widetilde{\nabla}_\theta f(\hat{\theta}_t)$ $\qquad\qquad\qquad\qquad\qquad\qquad$ ◇ Compute STE for compressed model
5: $\quad m_t = \beta_1 m_{t-1} + (1 - \beta_1)g_t$ $\qquad\qquad\qquad$ ◇ Update first-order gradient momentum
6: $\quad v_t = \beta_2 v_{t-1} + (1 - \beta_2)g_t^2$ $\qquad\qquad\qquad$ ◇ Update second-order gradient momentum
7: $\quad \tilde{v}_t = \max(v_t, \tilde{v}_{t-1})$ $\qquad\qquad\qquad\qquad\qquad$ ◇ Apply AMSGrad normalization
8: $\quad \theta_{t+1} = \theta_t - \eta \frac{m_t}{\sqrt{\tilde{v}_t} + \epsilon}$ $\qquad\qquad\qquad$ ◇ Update the uncompressed model parameters
9: **end for**

---

*Proof.* Let $G$ be the gradient bound with respect to $\ell_2$ norm, that is, $\|g_t\|_2 \leq G$. Using the relationship between $\ell_2$ and $\ell_\infty$ norms, we conclude $G \leq \sqrt{d}G_\infty$. Let $\Gamma_t = \text{Diag}^{-1/2}(\tilde{v}_t + \epsilon)$ be the preconditioning (diagonal) matrix and rewrite the main update rule as

$$\theta_{t+1} = \theta_t - \eta \Gamma_t m_t.$$

Letting $\theta_0 = \theta_1$, define virtual iterates $x_t$ as follows:

$$x_t = \frac{1}{1 - \beta_1}\theta_t - \frac{\beta_1}{1 - \beta_1}\theta_{t-1}.$$

In particular, $x_1 = \theta_1$. Then, the update rule for the virtual iterates becomes

$$
\begin{aligned}
x_{t+1} - x_t &= \frac{1}{1 - \beta_1}(\theta_{t+1} - \theta_t) - \frac{\beta_1}{1 - \beta_1}(\theta_t - \theta_{t-1}) \\
&= -\frac{\eta}{1 - \beta_1}\Gamma_t m_t + \frac{\eta\beta_1}{1 - \beta_1}\Gamma_{t-1}m_{t-1} \\
&= -\frac{\eta}{1 - \beta_1}\Gamma_t m_t + \frac{\eta\beta_1}{1 - \beta_1}\Gamma_{t-1}m_{t-1} \pm \frac{\eta\beta_1}{1 - \beta_1}\Gamma_t m_{t-1} \\
&= -\frac{\eta}{1 - \beta_1}\Gamma_t(m_t - \beta m_{t-1}) + \frac{\eta\beta_1}{1 - \beta_1}\underbrace{(\Gamma_{t-1} - \Gamma_t)}_{\overset{\text{def}}{=} \Delta\Gamma_t}m_{t-1} \\
&= -\eta\Gamma_t g_t + \frac{\eta\beta_1}{1 - \beta_1}\Delta\Gamma_t m_{t-1}.
\end{aligned}
$$

Next we apply smoothness (Assumption 1) of the loss function $f$ over the iterates $x_t$:

$$f(x_{t+1}) \leq f(x_t) + \langle \nabla f(x_t), x_{t+1} - x_t \rangle + \frac{L}{2}\|x_{t+1} - x_t\|^2.$$

Taking expectation and splitting the inner product into two part, we obtain

$$\mathbb{E}[f(x_{t+1})] - \mathbb{E}[f(x_t)]$$

$$\leq -\eta\mathbb{E}\left[\langle\nabla f(x_t), \Gamma_t g_t\rangle\right] + \eta\mathbb{E}\left[\left\langle\nabla f(x_t), \frac{\beta_1}{1-\beta_1}\Delta\Gamma_t m_{t-1}\right\rangle\right]$$

$$+ \frac{\eta^2 L}{2}\mathbb{E}\left[\left\|\Gamma_t g_t - \frac{\beta_1}{1-\beta_1}\Delta\Gamma_t m_{t-1}\right\|^2\right]$$

$$= \underbrace{-\eta\mathbb{E}\left[\langle\nabla f(\theta_t), \Gamma_t g_t\rangle\right]}_{I} + \underbrace{\eta\mathbb{E}\left[\left\langle\nabla f(x_t), \frac{\beta_1}{1-\beta_1}\Delta\Gamma_t m_{t-1}\right\rangle\right]}_{II}$$

$$+ \underbrace{\frac{\eta^2 L}{2}\mathbb{E}\left[\left\|\Gamma_t g_t - \frac{\beta_1}{1-\beta_1}\Delta\Gamma_t m_{t-1}\right\|^2\right]}_{III} + \underbrace{\eta\mathbb{E}\left[\langle\nabla f(\theta_t) - \nabla f(x_t), \Gamma_t g_t\rangle\right]}_{IV}. \quad (5)$$

In the following, we bound all the four terms mentioned above.

**Bounding term I.** Let $\|\Delta\Gamma_t\|$ be the operator norm (with respect to $\ell_2$ norm) of the matrix $\Delta\Gamma_t$. Since $\Delta\Gamma_t$ is diagonal, the spectral norm coincides with the largest diagonal value in magnitude. Using unbiasedness of the stochastic gradients, we have

$$I = -\eta\mathbb{E}\left[\langle\nabla f(\theta_t), \Gamma_{t-1}g_t\rangle\right] - \eta\mathbb{E}\left[\langle\nabla f(\theta_t), \Delta\Gamma_t g_t\rangle\right]$$

$$\leq -\eta\mathbb{E}\left[\left\langle\nabla f(\theta_t), \Gamma_{t-1}\nabla f(\widehat{\theta}_t)\right\rangle\right] + \eta G^2\mathbb{E}[\|\Delta\Gamma_t\|].$$

$$= -\eta\mathbb{E}\left[\left\langle\nabla f(\widehat{\theta}_t), \Gamma_{t-1}\nabla f(\widehat{\theta}_t)\right\rangle\right] + \eta\mathbb{E}\left[\left\langle\nabla f(\widehat{\theta}_t) - \nabla f(\theta_t), \Gamma_{t-1}\nabla f(\widehat{\theta}_t)\right\rangle\right] + \eta G^2\mathbb{E}[\|\Delta\Gamma_t\|].$$

$$\leq -\eta\lambda_{\min}(\Gamma_{t-1})\mathbb{E}[\|\nabla f(\widehat{\theta}_t)\|^2] + \eta LG\mathbb{E}[\|\Gamma_{t-1}\|\|\widehat{\theta}_t - \theta_t\|] + \eta G^2\mathbb{E}[\|\Delta\Gamma_t\|]$$

$$\leq -\frac{\eta}{C_0}\mathbb{E}[\|\nabla f(\widehat{\theta}_t)\|^2] + \eta LG\mathbb{E}[\|\widehat{\theta}_t - \theta_t\| \cdot \|\Gamma_{t-1}\|] + \eta G^2\mathbb{E}[\|\Delta\Gamma_t\|], \quad (6)$$

where we used Assumption 2 and Lemma 3 to bound

$$\lambda_{\min}(\Gamma_{t-1}) \geq (\|\tilde{v}_{t-1}\|_{\max} + \epsilon)^{-1/2} \geq (G^2 + \epsilon)^{-1/2} \stackrel{\text{def}}{=} \frac{1}{C_0}.$$

**Bounding term II.** Splitting the inner product again and bounded each term, we get

$$II = \eta\mathbb{E}\left[\left\langle\nabla f(\theta_t), \frac{\beta_1}{1-\beta_1}\Delta\Gamma_t m_{t-1}\right\rangle\right] + \eta\mathbb{E}\left[\left\langle\nabla f(x_t) - \nabla f(\theta_t), \frac{\beta_1}{1-\beta_1}\Delta\Gamma_t m_{t-1}\right\rangle\right]$$

$$\leq \frac{\eta\beta_1}{1-\beta_1}\mathbb{E}\left[\|\nabla f(\theta_t)\|\|\Delta\Gamma_t m_{t-1}\|\right] + \frac{\eta^2 L\beta_1^2}{(1-\beta_1)^2}\mathbb{E}\left[\|\Gamma_{t-1}m_{t-1}\| \cdot \|\Delta\Gamma_t m_{t-1}\|\right]$$

$$\leq \frac{\eta\beta_1}{1-\beta_1}G^2\mathbb{E}[\|\Delta\Gamma_t\|] + \frac{\eta^2\beta_1^2 LG^2}{(1-\beta_1)^2\sqrt{\epsilon}}\mathbb{E}[\|\Delta\Gamma_t\|], \quad (7)$$

where we used the fact that the largest eigenvalue $\lambda_{\max}(\Gamma_t) = \|\Gamma_t\| = (\|\tilde{v}_t\|_{\min} + \epsilon)^{-1/2} \leq \epsilon^{-1/2}$. The second inequality is due to the smoothness of $f$, and the last inequality is due to Lemma 1, Assumption 2 and the property of norms.

**Bounding term III.** This term can be bounded as follows:

$$III \leq \eta^2 L\mathbb{E}\left[\|\Gamma_t g_t\|^2\right] + \frac{\eta^2 L\beta_1^2}{(1-\beta_1)^2}\mathbb{E}\left[\|\Delta\Gamma_t m_{t-1}\|^2\right]$$

$$\leq \frac{\eta^2 L}{\epsilon}\mathbb{E}[\|g_t - \nabla f(\widehat{\theta}_t) + \nabla f(\widehat{\theta}_t)\|^2] + \frac{\eta^2 L\beta_1^2}{(1-\beta_1)^2}\mathbb{E}\left[\|\Delta\Gamma_t m_{t-1}\|^2\right]$$

$$\leq \frac{\eta^2 L}{\epsilon}\left(\mathbb{E}[\|\nabla f(\widehat{\theta}_t)\|^2] + \sigma^2\right) + \frac{\eta^2 L\beta_1^2 G^2}{(1-\beta_1)^2}\mathbb{E}[\|\Delta\Gamma_t\|^2]$$

$$\leq \frac{\eta^2 L}{\epsilon}\mathbb{E}[\|\nabla f(\widehat{\theta}_t)\|^2] + \frac{\eta^2 L\sigma^2}{\epsilon} + \frac{\eta^2 L\beta_1^2 G^2}{(1-\beta_1)^2}\mathbb{E}[\|\Delta\Gamma_t\|^2], \quad (8)$$

where we used Assumption 3 that $g_t$ is unbiased with bounded variance $\sigma^2$.

**Bounding term IV.** Finally, for the fourth term, we have

$$
\begin{aligned}
IV &= \eta\mathbb{E}\left[\langle\nabla f(\theta_t) - \nabla f(x_t), \Gamma_{t-1}g_t\rangle\right] + \eta\mathbb{E}\left[\langle\nabla f(\theta_t) - \nabla f(x_t), \Delta\Gamma_t g_t\rangle\right] \\
&\leq \eta\mathbb{E}\left[\left\langle\nabla f(\theta_t) - \nabla f(x_t), \Gamma_{t-1}\nabla f(\widehat{\theta}_t)\right\rangle\right] + \frac{\eta^2 L\beta_1}{1-\beta_1}\mathbb{E}\left[\|\Gamma_t m_{t-1}\|\,\|\Delta\Gamma_t g_t\|\right] \\
&\overset{(a)}{\leq} \frac{\eta\rho}{2\epsilon}\mathbb{E}[\|\nabla f(\widehat{\theta}_t)\|^2] + \frac{\eta}{2\rho}\mathbb{E}[\|\nabla f(\theta_t) - \nabla f(x_t)\|^2] + \frac{\eta^2\beta_1 LG^2}{(1-\beta_1)\sqrt{\epsilon}}\mathbb{E}[\|\Delta\Gamma_t\|] \\
&\overset{(b)}{\leq} \frac{\eta\rho}{2\epsilon}\mathbb{E}[\|\nabla f(\widehat{\theta}_t)\|^2] + \frac{\eta^3\beta_1^2 L^2}{2(1-\beta_1)^2\rho}\mathbb{E}\left[\|\Gamma_t m_{t-1}\|^2\right] + \frac{\eta^2\beta_1 LG^2}{(1-\beta_1)\sqrt{\epsilon}}\mathbb{E}[\|\Delta\Gamma_t\|] \\
&\leq \frac{\eta\rho}{2\epsilon}\mathbb{E}[\|\nabla f(\widehat{\theta}_t)\|^2] + \frac{\eta^3\beta_1^2 L^2}{2(1-\beta_1)^2\rho\epsilon}\mathbb{E}\left[\|m_{t-1}\|^2\right] + \frac{\eta^2 L\beta_1 G^2}{(1-\beta_1)\sqrt{\epsilon}}\mathbb{E}[\|\Delta\Gamma_t\|], \qquad (9)
\end{aligned}
$$

where (a) is due to Young's inequality and (b) is based on Assumption 1. Now integrating (6), (7), (8), (9) into (5),

$$
\begin{aligned}
I &\leq -\frac{\eta}{C_0}\mathbb{E}[\|\nabla f(\widehat{\theta}_t)\|^2] + \eta LG\mathbb{E}[\|\widehat{\theta}_t - \theta_t\|\cdot\|\Gamma_{t-1}\|] + \eta G^2\mathbb{E}[\|\Delta\Gamma_t\|] \\
II &\leq \frac{\eta\beta_1}{1-\beta_1}G^2\mathbb{E}[\|\Delta\Gamma_t\|] + \frac{\eta^2\beta_1^2 LG^2}{(1-\beta_1)^2\sqrt{\epsilon}}\mathbb{E}[\|\Delta\Gamma_t\|] \\
III &\leq \frac{\eta^2 L}{\epsilon}\mathbb{E}[\|\nabla f(\widehat{\theta}_t)\|^2] + \frac{\eta^2 L\sigma^2}{\epsilon} + \frac{\eta^2\beta_1^2 LG^2}{(1-\beta_1)^2}\mathbb{E}[\|\Delta\Gamma_t\|^2] \\
IV &\leq \frac{\eta\rho}{2\epsilon}\mathbb{E}[\|\nabla f(\widehat{\theta}_t)\|^2] + \frac{\eta^3\beta_1^2 L^2}{2(1-\beta_1)^2\rho\epsilon}\mathbb{E}\left[\|m_{t-1}\|^2\right] + \frac{\eta^2 L\beta_1 G^2}{(1-\beta_1)\sqrt{\epsilon}}\mathbb{E}[\|\Delta\Gamma_t\|],
\end{aligned}
$$

and taking the telescoping summation over $t = 1, \ldots, T$, we obtain

$$
\begin{aligned}
&\mathbb{E}[f(x_{T+1})] - \mathbb{E}[f(x_1)] \\
&\leq \left(-\frac{\eta}{C_0} + \frac{\eta^2 L}{\epsilon} + \frac{\eta\rho}{2\epsilon}\right)\sum_{t=1}^{T}\mathbb{E}[\|\nabla f(\widehat{\theta}_t)\|^2] + \frac{T\eta^2 L\sigma^2}{\epsilon} + \frac{\eta^3\beta_1^2 L^2}{2(1-\beta_1)^2\rho\epsilon}\sum_{t=1}^{T}\mathbb{E}\left[\|m_{t-1}\|^2\right] \\
&\quad + \left(\frac{\eta G^2}{1-\beta_1} + \frac{\eta^2\beta_1 LG^2}{(1-\beta_1)^2\sqrt{\epsilon}}\right)\sum_{t=1}^{T}\mathbb{E}[\|\Delta\Gamma_t\|] + \frac{\eta^2\beta_1^2 LG^2}{(1-\beta_1)^2}\sum_{t=1}^{T}\mathbb{E}[\|\Delta\Gamma_t\|^2] + \frac{\eta LG}{T}\sum_{t=1}^{T}\mathbb{E}[\|\widehat{\theta}_t - \theta_t\|\cdot\|\Gamma_{t-1}\|] \\
&\leq \left(-\frac{\eta}{C_0} + \frac{\eta^2 L}{\epsilon} + \frac{\eta\rho}{2\epsilon} + \frac{\eta^3\beta_2^2 L^2}{2(1-\beta_1)^2\rho\epsilon}\right)\sum_{t=1}^{T}\mathbb{E}[\|\nabla f(\widehat{\theta}_t)\|^2] + \frac{T\eta^2 L\sigma^2}{\epsilon} + \frac{T\eta^3 L^2\beta_1^2\sigma^2}{2(1-\beta_1)^2\rho\epsilon} \\
&\quad + \left(\frac{\eta G^2}{1-\beta_1} + \frac{\eta^2\beta_1 LG^2}{(1-\beta_1)^2\sqrt{\epsilon}}\right)\sum_{t=1}^{T}\mathbb{E}[\|\Delta\Gamma_t\|] + \frac{\eta^2\beta_1^2 LG^2}{(1-\beta_1)^2}\sum_{t=1}^{T}\mathbb{E}[\|\Delta\Gamma_t\|^2] + \frac{\eta LG}{T}\sum_{t=1}^{T}\mathbb{E}[\|\widehat{\theta}_t - \theta_t\|\cdot\|\Gamma_{t-1}\|],
\end{aligned}
$$

where we used Lemma 1. Choosing $\rho = \frac{\epsilon}{2C_0}$ and $\eta \leq \eta_0 \overset{\text{def}}{=} \frac{\epsilon(1-\beta_1)}{4LC_0}$ and using Lemma 2, we get

$$
\begin{aligned}
\mathbb{E}[f(x_{T+1}) - f(x_1)] &\leq -\frac{\eta}{2C_0}\sum_{t=1}^{T}\mathbb{E}[\|\nabla f(\widehat{\theta}_t)\|^2] + \frac{T\eta^2 L\sigma^2}{\epsilon} + \frac{T\eta^3 L^2 C_0\beta_1^2\sigma^2}{(1-\beta_1)^2\epsilon^2} \\
&\quad + \frac{2\eta G^2}{(1-\beta_1)\sqrt{\epsilon}} + \frac{4\eta^2\beta_1 LG^2}{(1-\beta_1)^2\epsilon} + \frac{\eta LG}{T}\sum_{t=1}^{T}\mathbb{E}[\|\widehat{\theta}_t - \theta_t\|\cdot\|\Gamma_{t-1}\|].
\end{aligned}
$$

Re-arranging terms, we get

$$\frac{1}{T}\sum_{t=1}^{T}\mathbb{E}[\|\nabla f(\widehat{\theta}_t)\|^2] \leq 2C_0\left(\frac{f(\theta_1)-f^*}{T\eta}+\frac{\eta L\sigma^2}{\epsilon}+\frac{\eta^2 L^2 C_0\beta_1^2\sigma^2}{(1-\beta_1)^2\epsilon^2}\right)$$

$$+4C_0\left(\frac{G^2}{T(1-\beta_1)\sqrt{\epsilon}}+\frac{\eta\beta_1 LG^2}{T(1-\beta_1)^2\epsilon}\right)+\frac{2C_0LG}{T}\sum_{t=1}^{T}\mathbb{E}\left[\frac{\|\widehat{\theta}_t-\theta_t\|_2}{\sqrt{\epsilon+\|\tilde{v}_{t-1}\|_{\min}}}\right],$$

where in the last inequality we used $x_1=\theta_1$ and the lower bound $f^*\leq f(\theta)$ for all $\theta\in\mathbb{R}^d$. Finally, choosing $\eta=\min(\eta_0,\frac{1}{\sqrt{T}})$ and considering the two cases, we continue

$$\frac{1}{T}\sum_{t=1}^{T}\mathbb{E}[\|\nabla f(\widehat{\theta}_t)\|^2] \leq 2C_0\left(\max\left(1,\frac{1}{\eta_0\sqrt{T}}\right)\frac{f(\theta_1)-f^*}{\sqrt{T}}+\frac{L\sigma^2}{\epsilon\sqrt{T}}+\frac{L^2C_0\beta_1^2\sigma^2}{(1-\beta_1)^2\epsilon^2 T}\right)$$

$$+4C_0\left(\frac{G^2}{T(1-\beta_1)\sqrt{\epsilon}}+\frac{\beta_1 LG^2}{T^{3/2}(1-\beta_1)^2\epsilon}\right)+\frac{2C_0LG}{T}\sum_{t=1}^{T}\mathbb{E}\left[\frac{\|\widehat{\theta}_t-\theta_t\|_2}{\sqrt{\epsilon+\|\tilde{v}_{t-1}\|_{\min}}}\right]$$

$$\leq 2C_0\left(\frac{f(\theta_1)-f^*}{\sqrt{T}}+\frac{L\sigma^2}{\epsilon\sqrt{T}}+\frac{L^2C_0\beta_1^2\sigma^2}{(1-\beta_1)^2\epsilon^2 T}\right)$$

$$+4C_0\left(\frac{f(\theta_1)-f^*}{2\eta_0 T}+\frac{G^2}{T(1-\beta_1)\sqrt{\epsilon}}+\frac{\beta_1 LG^2}{T^{3/2}(1-\beta_1)^2\epsilon}\right)$$

$$+\frac{2C_0LG}{\sqrt{\epsilon}}\mathbb{E}\left[\frac{1}{T}\sum_{t=1}^{T}\|\widehat{\theta}_t-\theta_t\|_2\right],$$

Using the bounds $G\leq\sqrt{N}G_\infty$, $C_0\leq\frac{\sqrt{N}}{2}C$ and surpessing higher order terms, we simplify the bound to

$$\frac{1}{T}\sum_{t=1}^{T}\mathbb{E}[\|\nabla f(\widehat{\theta}_t)\|^2]\leq\frac{CLG_\infty}{\sqrt{\epsilon}}\mathbb{E}\left[\frac{1}{T}\sum_{t=1}^{T}\|\widehat{\theta}_t-\theta_t\|_2\right]\cdot N+\frac{C\sqrt{N}}{\sqrt{T}}\left(f(\theta_1)-f^*+\frac{L\sigma^2}{\epsilon}\right)+\mathcal{O}\left(\frac{N^{3/2}}{T}\right),$$

which completes the proof of the theorem. $\qquad\square$

**Lemma 1.** *For any $t\geq 1$ the following bounds hold:*

$$\|m_t\|\leq G,\quad \sum_{t=1}^{T}\mathbb{E}\left[\|m_t\|^2\right]\leq T\sigma^2+\sum_{t=1}^{T}\mathbb{E}\left[\|\nabla f(\widehat{\theta}_t)\|^2\right]\tag{10}$$

*Proof.* Let us start with the proof of the first bound on $m_t$.

$$\begin{aligned}\|m_{t+1}\|^2 &= \|\beta_1 m_t+(1-\beta_1)g_{t+1}\|^2\\ &\leq \beta_1\|m_t\|^2+(1-\beta_1)\|g_{t+1}\|^2\\ &\leq \beta_1^t\|m_1\|+(1-\beta_1)\sum_{\tau=2}^{t+1}\beta_1^{t+1-\tau}\|g_\tau\|^2=(1-\beta_1)\sum_{\tau=1}^{t+1}\beta_1^{t+1-\tau}\|g_\tau\|^2.\end{aligned}$$

Using the bounded gradient assumption, we get

$$\|m_t\|^2\leq(1-\beta_1)G^2\sum_{\tau=1}^{t}\beta_1^{t-\tau}\leq G^2.$$

To derive the bound with expectation, we apply Cauchy-Schwartz inequality and the bounded variance assumption:

$$\sum_{t=1}^{T} \mathbb{E}\left[\|m_t\|^2\right] \leq (1 - \beta_1) \sum_{t=1}^{T} \sum_{\tau=1}^{t} \beta_1^{t-\tau} \mathbb{E}\left[\|g_\tau\|^2\right]$$

$$\leq \sum_{t=1}^{T} \mathbb{E}\left[\|g_t\|^2\right] = \sum_{t=1}^{T} \mathbb{E}\left[\|g_t - \nabla f(\widehat{\theta}_t) + \nabla f(\widehat{\theta}_t)\|^2\right]$$

$$\leq \sum_{t=1}^{T} \left(\sigma^2 + \mathbb{E}\left[\|\nabla f(\widehat{\theta}_t)\|^2\right]\right) = T\sigma^2 + \sum_{t=1}^{T} \mathbb{E}\left[\|\nabla f(\widehat{\theta}_t)\|^2\right].$$

$\square$

**Lemma 2.** *For* $\Delta\Gamma_t = \Gamma_{t-1} - \Gamma_t$ *we have*

$$\sum_{t=1}^{T} \|\Delta\Gamma_t\| \leq \frac{1}{\sqrt{\epsilon}}, \quad \sum_{t=1}^{T} \|\Delta\Gamma_t\|^2 \leq \frac{1}{\epsilon}.$$

*Proof.* From the definitions of $\Gamma_t = \mathrm{Diag}^{-1/2}(\tilde{v}_t + \epsilon)$ and $\tilde{v}_t = \max(v_t, \tilde{v}_{t-1})$ imply that $\Delta\Gamma_t = \Gamma_{t-1} - \Gamma_t$ is positive semidefinite. Hence, $\|\Delta\Gamma_t\| = \lambda_{\max}(\Delta\Gamma_t) \geq 0$. Using the convexity of $\lambda_{\max}$ over symmetric matrices, we get

$$\sum_{t=1}^{T} \|\Delta\Gamma_t\| = \max_i \sum_{t=1}^{T} \Delta\Gamma_{t,i}$$

$$= \max_i \sum_{t=1}^{T} \left(\frac{1}{\sqrt{\tilde{v}_{t-1,i} + \epsilon}} - \frac{1}{\sqrt{\tilde{v}_{t,i} + \epsilon}}\right) = \max_i \left(\frac{1}{\sqrt{\tilde{v}_{0,i} + \epsilon}} - \frac{1}{\sqrt{\tilde{v}_{T,i} + \epsilon}}\right) \leq \frac{1}{\sqrt{\epsilon}}$$

For the second sum of squared norms, notice that for scalars $a \geq b \geq 0$, it holds that

$$(a - b)^2 \leq (a - b)(a + b) = a^2 - b^2.$$

Therefore, the above derivation can be repeated without the square roots as follows:

$$\sum_{t=1}^{T} \|\Delta\Gamma_t\|^2 = \max_i \sum_{t=1}^{T} \Delta\Gamma_{t,i}^2$$

$$= \max_i \sum_{t=1}^{T} \left(\frac{1}{\sqrt{\tilde{v}_{t-1,i} + \epsilon}} - \frac{1}{\sqrt{\tilde{v}_{t,i} + \epsilon}}\right)^2$$

$$= \max_i \sum_{t=1}^{T} \left(\frac{1}{\tilde{v}_{t-1,i} + \epsilon} - \frac{1}{\tilde{v}_{t,i} + \epsilon}\right) = \max_i \left(\frac{1}{\tilde{v}_{0,i} + \epsilon} - \frac{1}{\tilde{v}_{T,i} + \epsilon}\right) \leq \frac{1}{\epsilon},$$

which completes the proof. $\square$

**Lemma 3.** *For all iterates* $t \geq 1$ *the following bound holds*

$$\|\tilde{v}_t\|_\infty \leq G^2.$$

*Proof.* From the update rules we get the bound for $v_t$ using the initialization $v_0 = 0$:

$$\|v_t\|_\infty \leq \|v_t\|_1 \leq \beta_2 \|v_{t-1}\|_1 + (1 - \beta_2)\|g_t\|^2$$

$$\leq \beta_2 \|v_{t-1}\|_1 + (1 - \beta_2)G^2$$

$$\leq \beta_2^t \|v_0\|_1 + (1 - \beta_2)G^2 \sum_{\tau=0}^{t-1} \beta_2^\tau \leq G^2.$$

Hence, using the update rule of $\tilde{v}_t$ and initialization $\tilde{v}_0 = 0$, we conclude

$$\|\tilde{v}_t\|_\infty \leq \max(\|v_t\|_\infty, \|\tilde{v}_{t-1}\|_\infty) \leq G^2.$$

$\square$

Next, we simplify the optimization setup by considering SGD optimizer over (still generally non-convex) quadratics. In this special case, we provide improved and generally optimal asymptotic convergence rate. Moreover, we do not use the bounded gradient condition (i.e., $\|g_t\|_\infty \le G_\infty$) of Assumption 2 in this analysis.

More formally, consider iterates $\theta_{t+1} = \theta_t - \eta \widetilde{\nabla}_\theta f(\widehat{\theta}_t)$, where $\widehat{\theta}_t = \mathcal{C}(\theta_t)$ is the compressed model. Suppose that the loss function is quadratic with Hessian matrix $\mathbf{H} \in \mathbb{R}^{N \times N}$ and our compression scheme $\mathcal{C} \colon \mathbb{R}^N \to \mathbb{R}^N$ is unbiased, namely $\mathbb{E}_{\mathcal{C}}[\widehat{\theta}_t] = \theta_t$. Since the loss is quadratic, we have

$$\nabla f(\widehat{\theta}_t) = \nabla f(\theta_t + (\widehat{\theta}_t - \theta_t)) = \nabla f(\theta_t) + \mathbf{H}(\widehat{\theta}_t - \theta_t).$$

Denote by $\mathbb{E}_t = \mathbb{E}[\cdot|\theta_t]$ the conditional expectation conditioned on iterate $\theta_t$, and apply unbiasedness of the compression to get

$$\mathbb{E}_t \|\nabla f(\widehat{\theta}_t)\|^2 = \|\nabla f(\theta_t)\|^2 + \mathbb{E}_t \|\mathbf{H}(\widehat{\theta}_t - \theta_t)\|^2 \tag{11}$$

Therefore,

$$
\begin{aligned}
&\mathbb{E}_t[f(\theta_{t+1}) - f^*] \\
\le\ & (f(\theta_t) - f^*) - \eta \mathbb{E}_t[\langle \nabla f(\theta_t), \widetilde{\nabla} f(\widehat{\theta}_t)\rangle] + \frac{L\eta^2}{2}\mathbb{E}_t[\|\widetilde{\nabla} f(\widehat{\theta}_t)\|^2] \\
\le\ & (f(\theta_t) - f^*) - \eta \mathbb{E}_t[\langle \nabla f(\theta_t), \nabla f(\widehat{\theta}_t)\rangle] + \frac{L\eta^2}{2}\mathbb{E}_t[\|\nabla f(\widehat{\theta}_t)\|^2] + \frac{L\eta^2}{2}\sigma^2 \\
=\ & (f(\theta_t) - f^*) - \eta \mathbb{E}_t[\|\nabla f(\theta_t)\|^2] + \frac{L\eta^2}{2}\mathbb{E}_t[\|\nabla f(\theta_t)\|^2] + \frac{L\eta^2}{2}\mathbb{E}_t[\|\mathbf{H}(\widehat{\theta}_t - \theta_t)\|^2] + \frac{L\eta^2}{2}\sigma^2 \\
=\ & (f(\theta_t) - f^*) - \eta(1 - L\eta/2)\mathbb{E}_t[\|\nabla f(\theta_t)\|^2] + \frac{L\eta^2}{2}(\mathbb{E}_t[\|\widehat{\theta}_t - \theta_t\|^2_{\mathbf{H}^2}] + \sigma^2) \\
\le\ & (f(\theta_t) - f^*) - \frac{\eta}{2}\mathbb{E}_t[\|\nabla f(\theta_t)\|^2] + \frac{L\eta^2}{2}(\mathbb{E}_t[\|\widehat{\theta}_t - \theta_t\|^2_{\mathbf{H}^2}] + \sigma^2),
\end{aligned}
$$

where we used $\mathbb{E}_t[\nabla f(\widehat{\theta}_t)] = \nabla f(\theta_t)$ due to the unbiasedness of compression and enforced the bound $\eta \le \frac{1}{L}$ in the last step. Hence,

$$\frac{1}{T}\sum_{t=1}^T \mathbb{E}[\|\nabla f(\theta_t)\|^2] \le \frac{2(f(x_1) - f^*)}{\eta T} + \eta L \left( \mathbb{E}\left[\frac{1}{T}\sum_{t=1}^T \|\widehat{\theta}_t - \theta_t\|^2_{\mathbf{H}^2}\right] + \sigma^2 \right).$$

Choosing the step size $\eta = \min(\frac{1}{L}, \frac{1}{\sqrt{T}})$ and applying $L$-smoothness, we get $\mathcal{O}(1/\sqrt{T})$ convergence rate for the uncompressed iterates $\theta_t$:

$$\frac{1}{T}\sum_{t=1}^T \mathbb{E}[\|\nabla f(\theta_t)\|^2] \le \frac{1}{\sqrt{T}}\left( 2(f(x_1) - f^*) + L\sigma^2 + L^3 \mathbb{E}\left[\frac{1}{T}\sum_{t=1}^T \|\widehat{\theta}_t - \theta_t\|^2_2\right]\right) \max\left(1, \frac{L}{\sqrt{T}}\right).$$

For the convergence bound with respect to the compressed iterates $\widehat{\theta}_t$, we apply (11) to quantify the exact difference in average gradient norms with the following identity:

$$\frac{1}{T}\sum_{t=1}^T \mathbb{E}[\|\nabla f(\widehat{\theta}_t)\|_2^2] = \frac{1}{T}\sum_{t=1}^T \mathbb{E}[\|\nabla f(\theta_t)\|_2^2] + \mathbb{E}\left[\frac{1}{T}\sum_{t=1}^T \|\mathbf{H}(\widehat{\theta}_t - \theta_t)\|_2^2\right].$$

Thus, a randomly chosen compressed iterate $\widehat{\theta}$ from $\{\widehat{\theta}_1, \dots, \widehat{\theta}_T\}$ satisfies

$$\mathbb{E}[\|\nabla f(\widehat{\theta})\|^2] \le L^2 \cdot \mathbb{E}\left[\frac{1}{T}\sum_{t=1}^T \|\widehat{\theta}_t - \theta_t\|_2^2\right] + \mathcal{O}\left(\frac{1}{\sqrt{T}}\right).$$

# F Improved Sparse Training via RBBM

## F.1 Comparison against RigL

In this subsection we compare our backward mask heuristic in Figure5d with the RigL method of (Evci et al., 2020). We run two instances of RigL: 1) the default one that updates the mask once at

100 steps (i.e. $\Delta = 100$) and updates the mask for the last time at 75% of training and 2) a version of RigL that is closer to our RBBM setup, which changes the mask at each step (i.e. $\Delta = 1$) during the entire training. In Figure 11a we observe that both versions of RigL induce lower capacity than our naive baseline for a fixed sparsity.

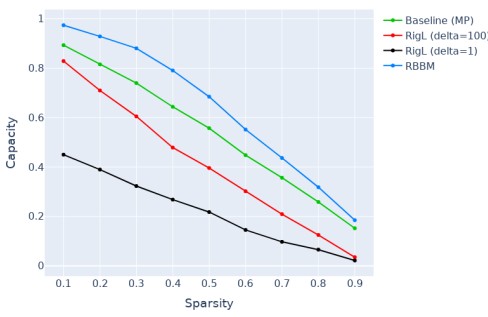 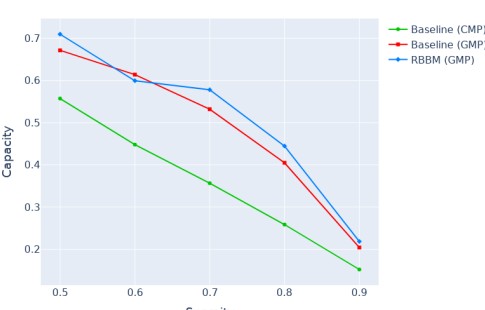

(a) Pre-training Llama-30M with different sparsities using our MP baseline, RBBM heuristic and RigL variations.

(b) Pre-training Llama-30M with different sparsities using our constant MP (CMP) baseline, GMP and RBBM heuristic with GMP schedule.

Figure 11: Comparison of sparse training methods for Llama-30M.

## F.2 Comparison against Gradual Magnitude Pruning (GMP)

In this section we show our results for applying the GMP sparsity schedule [37] for our setup in Figure 11b. Our first baseline is the constant Magnitude Pruning (CMP), where the backward mask is identical to the forward mask (determined by Top-K) and the sparsity is kept fixed during training. The second baseline is the original GMP where sparsity increases gradually and we compare against the gradual sparsity schedule applied to our **b-rms** heuristic.

We observe our RBBM heuristic with GMP schedule has lower capacity than both CMP and GMP baselines when sparsity is $< 40\%$. However, for sparsities $\geq 40\%$ there is no significant difference in capacity between CMP and GMP schedules.

## F.3 Backward Mask Heuristics

In this section we provide more details about our backward heuristics.

Our purpose is to perform sparse training for both forward and backward passes. All models are trained with the same learning rates as in the Quest project.

**Notations.** Let $\theta$ be the model parameters, $M_{FW}$ be the be the mask for the forward pass, and $M_{BW}$ be the mask for the backward pass.

**Forward Pass.** The mask for the forward pass is computed using Top-K operator, where $K$ is chosen based on the target sparsity. Supposing Top-K returns the indices of largest entries by magnitude, the $i^{th}$ entry in the forward mask for a tensor $x$ is computed using the indicator function $\mathbb{I}$ as follows:

$$M_{FW}^i(x) = \mathbb{I}[i \in TopK(|x|)] \tag{12}$$

**Backward Pass.** The mask for the backward pass is computed using a few heuristics described below.

1. **fw:** the backward mask is simply set to the forward mask: $M_{BW} = M_{FW}$. This heuristic allows gradients to flow only through the largest parameters by magnitude selected by Top-K, while the low-magnitude parameters will have zero gradient

2. **rms:** we align the tensor $x$ with the standard normal distribution by dividing $x$ by its root mean square $RMS(x) = \sqrt{\frac{1}{n} \sum_{i=1}^{n} x_i^2}$, which results in $||x/RMS(x)||_2^2 = n$. For this heuristic, the

user sets a median deviation parameter $p \in (0, 0.5)$, which is used to determine the threshold for the backward mask $T_{RMS}(x, p) = RMS(x) \cdot ppf(0.5 + p)$, where $ppf$ is the inverse cumulative distribution function of the standard normal distribution (see the scipy.stats.norm.ppf function). The multiplication with $RMS(x)$ has the purpose of converting the threshold for the standard normal distribution to the threshold for the vector $x$. As a result, $M_{BW} = |x| > T_{RMS}(x, p)$.

3. **banded-rms (b-rms):** the **rms** heuristic has the property that the absolute values of $x$ that are larger than the threshold $T_{RMS}(x, p)$ will have value 1 in the mask, while the smaller ones will have value 0. This banded heuristic determines the backward mask using the threshold $T_{RMS}(x, p)$ computed for the **rms** heuristic in conjunction with the Top-K threshold (which we denote by $T_k$). We want to allow gradients to flow for the small parameters and create a band between $T_{RMS}(x, p)$ and $T_k$ where we do not allow gradients. Concretely, the backward mask is set as follows: $M_{BW} = (|x| < min(T_{RMS}(x, p), T_k)) \vee (max(T_{RMS}(x, p), T_k) < |x|)$. Since the median deviation $p$ is a hyper-parameter, we do not have any control over the relationship between $T_{RMS}(x, p)$ and $T_k$ and we are using the $min$ and $max$ functions to make sure the band is valid, e.g. the parameters do not receive gradient if they lie in the interval $[min(T_k, T_{RMS}(x, p)), max(T_k, T_{RMS}(x, p))]$.

4. **area-banded-rms (a-b-rms):** in the **b-rms** heuristic we do not have any control over the relationship between the Top-K threshold $T_k$ and $T_{RMS}(x, p)$. Let us discuss the two possible cases:

   (a) $T_k < T_{RMS}(x, p) : M_{BW} = (|x| < T_k) \vee (T_{RMS}(x, p) < |x|)$, which means that all values from $x$ with a lower magnitude than $T_k$ and larger magnitude than $T_{RMS}(x, p)$ will get gradient, while the values in the range $[T_k, T_{RMS}(x, p)]$ will not receive gradient, even though they were selected among the Top-K during the forward pass.

   (b) $T_{RMS}(x, p) < T_k : M_{BW} = (|x| < T_{RMS}(x, p)) \vee (T_k < |x|)$, which is the desired case we developed the **b-rms** heuristic for: the largest entries from $x$ according to the Top-K rule will receive gradient, as well as the entries smaller than $T_{RMS}(x, p)$. The entries lying in the interval $[T_{RMS}(x, p), T_k]$ will not receive gradient.

We want to make sure that case **(a)** above does not happen in practice and force the heuristic to behave as in the case **(b)**. For this, we need to change the way we compute the threshold $T_{RMS}(x, p)$.

The **area-b-rms** heuristic uses the area hyper-parameter $a \in [0, 1]$ (instead of median deviation $p$) and expresses the width of the band starting from the Top-K parameter $T_k$ towards zero to compute the threshold $T_a$ to make sure the condition $T_a < T_k$ always holds. As a result, $M_{BW} = (|x| < T_a) \vee (T_k < |x|)$. For example, $a = 0$ yields $T_k = T_a$ and this heuristic turns into **fw**, while $a = 1$ yields $T_a = 0$ and is equivalent to $M_{BW} = \mathbf{1}_d$ (all entries set to 1, meaning all parameters get gradients). When $a \in (0, 1)$, the parameters smaller than $T_a$ or larger than $T_k$ get gradients, while the parameters lying in the interval $[T_a, T_k]$ do not get gradients.

**How to compute the threshold $T_a$?** Compared to the threshold computation for the previous heuristic, the definition for $T_a$ is slightly more complicated and it was computed graphically. Let $f$ be the $cdf$ function and $f^{-1}$ be the $ppf$ function (inverse cdf) for the standard normal distribution.

$$T_a = RMS(x) \cdot f^{-1}\left(0.5 + (1 - a) \cdot \left(f\left(\frac{T_k}{RMS(x)}\right) - 0.5\right)\right) \tag{13}$$

**Explanations for the formula above.** Suppose the Top-K threshold $T_k$ has a corresponding cdf of 0.8 and we set $a = 0.5$ (which means 50%). We need to set the threshold $T_a$ such that $(f(T_k) - f(T_a))/(f(T_k) - 0.5) = a$, where 0.5 is the cdf of the mean (which is identical to the median for a standard normal distribution). This ratio expresses the length of the band $[0.5, f(T_k)]$ in the cdf space starting from $f(T_k)$ towards the median. As a consequence, the threshold $T_a = ppf(0.65)$ because the quantile 0.65 is the center of the interval $[0.5, cdf(T_k)] = [0.5, 0.8]$. The explanations of each term follow:

$$T_a = RMS(x) \cdot f^{-1} \underbrace{\left( 0.5 + \underbrace{(1-a) \cdot \underbrace{\left( \underbrace{f\left(\frac{T_k}{RMS(x)}\right)}_{=A} - 0.5 \right)}_{=B}}_{=C} \right)}_{=E}}_{=F}$$

- **A:** $f = cdf$ computes the corresponding quantile of the Top-K threshold $T_k$ normalized by $RMS(x)$
- **B:** subtract $0.5$ from term $A$ to compute the length of the interval $[0.5, f(T_k^{RMS})]$
- **C:** multiply by $1 - a$ because we take into consideration the band length that starts at $f(T_k^{RMS})$ towards $0$
- **D:** compute the cdf of $T_a$ by offsetting again by $0.5$ (the quantile of the median)
- **E:** use $ppf = f^{-1}$ to obtain the value that corresponds to $cdf(T_a)$ for the standard normal distribution
- **F:** multiply by $RMS(x)$ to obtain $T_a$ in the same space as $x$

**Technical note.** One could determine the threshold $T_a$ naively by employing the formula $T_a^{naive} = (1-a)T_k$. Despite simpler, this naive approach leads to a narrower band because the cdf space is non-linear.

**Conclusion.** The mask computed using the **a-b-rms** heuristic is more straightforward to understand because the parameter $a$ describes the area of the red band (where parameters do not receive gradients) as a percentage of the area between $0$ and the Top-K threshold $T_k$. This heuristic can be used as a replacement for **b-rms** and the parameter $a$ should be tuned, similarly to parameter $p$ for **b-rms**, with the distinction that $a \in [0, 1]$ (for **a-b-rms**) and $p \in (0, 0.5)$ (for **b-rms**).

