# OpenReview forum: "Unified Scaling Laws for Compressed Representations"
_NeurIPS.cc/2025/Conference — NeurIPS 2025 poster_

### Official Review · Reviewer_W7rD · 2025-06-19

**Clarity:** 3
**Significance:** 3
**Originality:** 3
**Rating:** 5
**Confidence:** 2

**Summary:**

This paper proposes a unified framework for compression scaling laws. It uses Gaussian Mean Square Error (GMSE) to define representation capacity, which helps predicting the loss for different compression techniques including quantization, sparsification, vector quantization, and their hybrids. Their representation capacity factorizes across different compression types. Applications include comparing different numeral formats and designing new methods (RBBM) for sparse training.

**Questions:**

1. Line 213: Could you explain why the noise injection gives a proxy for the ”information-theoretic lower bound”?
2. Is it possible to do experiments on different D/N ratios, even just on small N? If your scaling law holds, should the losses decrease at the same speed w.r.t. D, for different compression settings?

Typos:

line 217: shiowing —> showing

line 230: M:,0 —> M_{:,0}

**Ethical Concerns:**

["NO or VERY MINOR ethics concerns only"]

**Final Justification:**

The author addressed all my questions in the rebuttal. I think the additional experiments are convincing.

**Limitations:**

yes

**Quality:**

3

**Strengths And Weaknesses:**

Strengths:
1. The paper provides a simple and performant compression scaling law, generalizing all previous related works such that the prediciton can be done over a combination of compression techniques.
2. The experiments for factorization of representation capacity into up to 3 terms are a highlight point.

Weaknesses:
1. All models were trained up to a maximum of 200M parameters. This is relatively small compared with other papers on scaling law such as [1] (1.7B). The reliability of extrapolating the law to billion parameter regime is to be explored.
2. The experiments are based on a single architectural familty (Llama-style decoder-only) and a fixed D/N ratio (100toks/params).
3. The writing could be clearer. For example the representation R is not defined. Do you mean it the same as \hat{\theta_t}?

[1] Kumar, Tanishq, et al. "Scaling laws for precision."

---

> ### Author Rebuttal · Authors · 2025-07-30
>
> Thank you for your feedback!
>
> **1. On generalization to billion parameters.**
>
> Our original experimental design is very similar to those of prior work on compression scaling laws, e.g. Kumar et al.
> Nevertheless, to address your question, we trained an additional 1.6B‑parameter LLaMA‑family model with 4‑bit weight and activation quantization on 160B C4 tokens. The table below shows the observed C4 validation loss in the D/N=100 regime, as well as the prediction from the Decoupled scaling law fitted on models up to 200M parameters.
>
> | N    | Predicted Loss | Real Loss |
> |------|----------------|-----------|
> | 430M | 2.627          | 2.614     |
> | 800M | 2.523          | 2.495     |
> | 1.6B | 2.421          | 2.378     |
>
> Notably, the prediction errors are ~2x smaller than the gaps in performance between adjacent models even for models as large as the whole order of magnitude above those used for scaling law fitting.
> Additionally, we’d like to note that Kumar et al. did not properly scale to 1.6B parameters, as their 1.6B model was heavily undertrained at D/N=10, way below the compute optimality ratio and their main explored data saturation regions.
>
> **2. Generalization beyond LLaMa on C4.**
>
> We agree it is important to demonstrate that our findings are not specific to the LLaMa/C4 setting. To address this, we now present a series of experiments in an alternative training setup:
>  - Architecture: two‑layer ReLU² MLPs (vs three-layer scheme for LLaMa), RMSNorm applied after each sublayer (vs pre-normalization for LLaMa), and weight‑tied embeddings (vs untied for LLaMa).
>  - Dataset: high-quality NVIDIA ClimbMix dataset (vs C4 used previously).
>
> We conducted training with identical model sizes, data-saturation ratios, and optimizer hyperparameters. We refitted all the scaling law parameters from scratch.
> We found that the new estimates for parameters (α,β,E,F,L,C) match the LLaMa/C4 fit: their point estimates coincide within the confidence intervals, obtained by the bootstrapping procedure described in our paper.
>
> |      | LLaMa/C4 | Transformer/ClimbMix10M (new) |
> |:---------|:--------:|:---------:|
> | $\alpha$  | $0.13 \pm 0.05$  | $0.18 \pm 0.03$  |
> | $\beta$  | $0.33 \pm 0.06$ | $0.26 \pm 0.02$  |
> | E  | $1.3 \pm 0.5$ |  $1.4 \pm 0.3$  |
> | L | $1.0 \pm 0.7$ |  $0.84 \pm 0.1$  |
> | F | $0.41 \pm 0.02$  | $0.37 \pm 0.04$  |
> | C  | $1.39 \pm 0.08$  | $1.24 \pm 0.13$  |
>
> This indicates that the same underlying law governs both model families and datasets. We will add those results in the revised version of the paper. We thank the reviewer for the suggestion.
>
>
> **3. Definition of the representation.**
>
> Thank you for bringing this up, please see the definition below:
>
> Def. Representation \textbf{R}: a functional mapping of a full-precision tensor $W$ (for instance, BF16) to a compressed format $R(W)$ (e.g. quantization, sparsification, etc.). The mapping may be point-wise (e.g., scalar quantization) or group-wise (e.g., vector quantization).
>
> We will add a complete definition in the revised version of the paper.
>
>
> **4. On the theoretic lower bound.**
>
> See the lower bound (13.24) in Theorem 13.3.2 of Cover, T.M. and Thomas, J.A. (2006), Elements of Information Theory, John Wiley & Sons, Inc., Hoboken. For a given compression rate or bit-rate the MSE is lower bounded by $\sigma^2 \cdot 4^{-\text{bit-rate}}$.
>
>
> **5. D/N ratios.**
>
> We apologize if it wasn't clearly stated in the text, we conducted the experiments for all models on a set of D/N ratios = [50, 100, 200].

---

> > ### Comment · Reviewer_W7rD · 2025-08-04
> >
> > Thank you for the detailed rebuttal. My questions are clearly addressed and I therefore raise my score.

---

### Official Review · Reviewer_4EjT · 2025-06-28

**Clarity:** 3
**Significance:** 4
**Originality:** 3
**Rating:** 5
**Confidence:** 4

**Summary:**

This paper introduces a unified scaling law that models the performance of compressed neural network representations, including sparsity, quantization, and their hybrid combinations.
To generalize across compression methods, the authors propose a novel metric called representation capacity $\rho$, which is independent of specific compression schemes and avoids reliance on traditional metrics like bit-rate or sparsity level.
This capacity $\rho$ is quantitatively defined by a compressed model's ability to fit random Gaussian data, measured using Gaussian Mean Squared Error (GMSE).
Using this representation capacity, the authors derive a generalized scaling law of the form:
${\rm Loss}(N,D) \sim A(N \rho(R))^{−\alpha} +BD^{-\beta} + E $, where $N$ is the number of model parameters, $D$, the data size, and $A, B, E, \alpha$ and $\beta$ are fitted constants
Furthermore, they show that the representation capacity of hybrid compression methods factorizes across components. For example, combining sparsity and quantization leads to:$\rho_{\rm total} = \rho_{\rm sparsification}*\rho_{\rm quantization}$.
The proposed scaling law and factorization principle are empirically validated across a wide range of compression schemes, including scalar quantization, vector quantization, sparse representations, and sparse-quantized models.

**Questions:**

1. The theoretical analysis in section 3 explains (very elegantly, I must say) the reasoning behind the RMSE contribution for the weight sparsification/quantization.
How does activation sparsification/quantization affects the loss. Why does it enter the formulation in Eq. 1 as a factor of $N$, the model weights.
will it hold also for convolutional networks, where $N$ can be small (i.e., convolution weights) while the activations might be much larger (i.e., image size).

2. Measuring of actual $\rho(R)$ values is not fully clear from the paper. I think I understand it as follows:
For each compression format $R$ you vary only $N$ (the model size) and then fit the training loss $L$ to the loss approximation, Eq. 1.
By fixing the data size $D$ you can avoid fitting $B$ and $\beta$ but rather only a total constant $E'=B*D^{-\beta}+E$.
Having a multitude of such experiments, per $R$, you fit the global parameters $A, \alpha$ and $E'$ as well as $\rho(R)$ per experiment.
I think what confused me is that you have 2 fit procedures -- one for Eq. 1 and then another one for Eq. 3.
Please verify in your responding to the review. In addition, whether I am right or wrong, the paper should clarify this methodology more elaborately.

3. Please put a refernece on how you obtain/calculate the theoretical lower bound (e.g., at 257). It might be also that I missed it...

4. In the checklist you mention "Code for reproducing the results is provided". However, a) you do not refer to it in the paper, b) I couldn't find such code in any link or supported material

5. How do you derive Eq. 3? This was unclear to me.
Later on I found some explanation in appendix D.2. Please refer to it from within the text.

6. small typos that I found. I can't avoid noting on it :)

line 217 shiowing -> showing

line 277 "the the" -> the

line 605 "the the" -> the

**Ethical Concerns:**

["NO or VERY MINOR ethics concerns only"]

**Final Justification:**

I thank the authors for their detailed reply and addressing of my concerns

I keep my score at 5

Comparing to scores provided by my peer reviewers strengthen my decision.

**Limitations:**

Should address limitation about the generality of the model to other architectures. see also comment above about activation quantization and CNN models

**Paper Formatting Concerns:**

The paper follows NeurIPS format

**Quality:**

3

**Strengths And Weaknesses:**

Strengths
- Novel unifying framework: Introduces a capacity-based scaling law that generalizes multiple prior works and applies to hybrid compressed formats, addressing a significant open problem in scaling compressed models.
- Strong theoretical grounding: Provides convergence analysis under Adam optimization using compressed iterates, connecting capacity with optimization behavior.
- Empirical validation: Demonstrates strong empirical fits across a wide variety of representations and model configurations (quantized, sparse, and their combinations).
- Practical relevance: The ability to rank compressed representations without retraining or exhaustive grid search is immediately useful for both research and deployment.
- Factorization finding: The observation that capacity factorizes across representation components is elegant and potentially impactful for compression-aware hyperparameter tuning.

Weaknesses
- The experimental scope is rather narrow and conducted on small Llama-style decoder-only models (up to 200M params) trained on the C4 dataset. Thus, it is not empirically shown how this law generalizes to other architectures (e.g., CNNs)
- The factorization of the representation capacity $\rho$ breaks at low bit rate, probably due to substantial second-order effects, as acknowledged by the authors.
However, there is no deep analysis where and why these breakdowns occur.

---

> ### Author Rebuttal · Authors · 2025-07-30
>
> Thank you for your feedback!
>
>
> ### Answers to your questions:
>
>
> **1. The theoretical analysis in section 3.**
>
>
> The theory in Section 3 indeed justifies RMSE reasoning for weight compression and does not consider activation/gradient compression.
>
>
> If we quantize/sparsify activations, then instead of mini-batch stochastic gradient $g_t$, we would get noisy gradients $\hat{g}_t$ with another level of noise. We can repeat the analysis of Theorem 1 and derive similar rate where RMSE term would be expressed in terms of gradient bias, namely $|| \hat{g}_t - g_t ||_2$ instead of $L|| \hat{\theta}_t - \theta_t ||_2$.
> However, it is important to note that $|| \hat{g}_t - g_t ||_2$ is not the compression RMSE of gradients since $\hat{g}_t$ is not the quantized/sparsified $Q(g_t)$ as it was the case for weight compression (activation compression affects gradient computation in a more complicated manner).
>
>
> **2. The fitting procedure.**
>
>
> Thanks for bringing this up, we need to clarify that:
>  - We always fit the parametric forms (e.g., Equation 3) on the validation loss prediction in the form of Equation 1 using the Huber loss.
>  - For the main setups of each compression format (143 runs for uniform quantization and noise injection, 250 runs for basic sparsity) we vary model sizes N, token count D and precision/sparsity to measure the final validation loss L as a function of N,D and compression degree to independently fit laws of forms from Table 1. Table 1 and Figure 1 present results based on these fits.
>  - For more focused additional experiments (Sections 4.3, 5.1, 5.2), we reuse the previously fitted general A,B,E,$\alpha$,$\beta$ parameters from the Decoupled scaling law fit and perform a pin-point fit of just the representation-related parameters. We do this to increase the fit robustness.
>
>
> **3. On the theoretical lower bound.**
>
>
> See the lower bound (13.24) in Theorem 13.3.2 of Cover, T.M. and Thomas, J.A. (2006), Elements of Information Theory, John Wiley & Sons, Inc., Hoboken. For a given compression rate or bit-rate the MSE is lower bounded by $\sigma^2 \cdot 4^{-\text{bit-rate}}$.
>
>
> **4. The codebase.**
>
>
> We apologize for the omission. Unfortunately we are not allowed to provide code links during the rebuttal, but all our code will be made open source and we will add a link to a public github repository in the revised version of the paper.
>
>
> **5. Cross-references and typos.**
>
>
> Thanks for catching this, we will make sure to refer to the appendix section in the revised version of the paper.
>
>
> ### Comments regarding the stated weaknesses:
> **1. Generalization beyond LLaMa on C4.**
>
>
> We agree it is important to demonstrate that our findings are not specific to the LLaMa/C4 setting. To address this, we now present a series of experiments in an alternative training setup:
>  - Architecture: two‑layer ReLU² MLPs (vs three-layer scheme for LLaMa), RMSNorm applied after each sublayer (vs pre-normalization for LLaMa), and weight‑tied embeddings (vs untied for LLaMa).
>  - Dataset: high-quality NVIDIA ClimbMix dataset (vs C4 used previously).
>
>
> We conducted training with identical model sizes, data-saturation ratios, and optimizer hyperparameters. We refitted all the scaling law parameters from scratch.
> We found that the new estimates for parameters (α,β,E,F,L,C) match the LLaMa/C4 fit: their point estimates coincide within the confidence intervals, obtained by the bootstrapping procedure described in our paper.
>
>
> |      | LLaMa/C4 | Transformer/ClimbMix10M (new) |
> |:---------|:--------:|:---------:|
> | $\alpha$  | $0.13 \pm 0.05$  | $0.18 \pm 0.03$  |
> | $\beta$  | $0.33 \pm 0.06$ | $0.26 \pm 0.02$  |
> | E  | $1.3 \pm 0.5$ |  $1.4 \pm 0.3$  |
> | L | $1.0 \pm 0.7$ |  $0.84 \pm 0.1$  |
> | F | $0.41 \pm 0.02$  | $0.37 \pm 0.04$  |
> | C  | $1.39 \pm 0.08$  | $1.24 \pm 0.13$  |
>
>
> This indicates that the same underlying law governs both model families and datasets. We will add those results in the revised version of the paper. We thank the reviewer for the suggestion.
>
>
> **2. On extreme sparsity.**
>
>
> We agree that separability breaks in the < 4‑bit regime. Namely, most optimizers' performance degrades, see for example Learned Step Size Quantization by K. Esser et al., where they address low-bit optimizer issues by modifying the training procedure (learned step sizes, gradient scaling, hyper-parameter tweaks).
> Also, kernel support for low bitwidth is limited  and prior work also shows similar degradation.
> These effects jointly account for the decoupling in this range, so we intentionally scoped our main claims to ≥4‑bit where the factorization is stable.
> We believe it should be possible to improve fit by focusing specifically on this low-bitwidth regime: currently, our laws span an extremely wide model size range.

---

> > ### Comment · Reviewer_4EjT · 2025-08-02
> >
> > Thank you for the detailed rebuttal answer.
> > Please incorporate the detailed explanations as well as the new results to the paper to make it clearer and to strengthen its validity.
> > I keep my score at "Accept"
> > Well Done!

---

### Official Review · Reviewer_ygts · 2025-07-02

**Clarity:** 4
**Significance:** 3
**Originality:** 2
**Rating:** 4
**Confidence:** 3

**Summary:**

The paper studies the scaling behavior of compressed models, unifying different ways and levels of sparsity and quantization. The proposed law can be used to predict representation capacity that enables comparing sparse, quantized, or both sparse and quantized models.

**Questions:**

Some questions are listed above.

**Ethical Concerns:**

["NO or VERY MINOR ethics concerns only"]

**Final Justification:**

I thank the authors for clarifying the connections to Kumar et al and Frantar et al. The additional experimental results also strengthen the claims of the paper. I increase my score to Borderline Accept.

**Limitations:**

Yes

**Paper Formatting Concerns:**

No issue detected

**Quality:**

2

**Strengths And Weaknesses:**

Strengths:
- The idea of factorizing capacity scores is very interesting.
- The proposed laws allows for predicting representation capacity in general, without restricting the space in only sparsity or quantization.

Weaknesses:
- The title in the PDF and the registered title do not match.
- While the capacity factorization is interesting, the high-level idea was introduced by Kumar et al.
- The paper has extensive results on Llama trained on C4. But it's unclear whether the findings and comparisons are generalized to other models and datasets.
- Looking at Table 1, Frantar et al [9] reaches smaller fit error for both sparsity and quantization compared to the proposed law in this paper. So, there is no definite gain over this baseline. Could there be a way to make better predictions using these two different laws?

---

> ### Author Rebuttal · Authors · 2025-07-30
>
> We thank the reviewer for their feedback!
>
>
> **1. Title mismatch.**
>
> We apologize and will fix this: the OpenReview title will be updated to match the PDF internal title.
>
>
> **2. Relation to Kumar et al.**
>
> Indeed, Kumar et al. introduced a notion of “representation capacity” in the context of quantization. Our contribution is different in three key ways:
>   1. Our functional form of representation capacity is more general, as it covers multiple forms of compression, e.g. sparsity and quantization. Further, it yields a much better fit in terms of prediction error, that is shown in Figure 1a, for example.
>   2. Even focusing on quantization, Kumar et al. only investigate scalar quantization with straight-through estimators, whereas we conducted experiments on formats including sparse + quantized, vector quantization, and multiple gradient estimators (STE, LSQ, QuEST).
>   3. We are the first to provide theoretical evidence for the parameter dependency being a function of the MSE.
>
> **3. Generalization beyond LLaMa on C4.**
>
> We agree it is important to demonstrate that our findings are not specific to the LLaMa/C4 setting. To address this, we now present a series of experiments in an alternative training setup:
>  - Architecture: two‑layer ReLU² MLPs (vs three-layer scheme for LLaMa), RMSNorm applied after each sublayer (vs pre-normalization for LLaMa), and weight‑tied embeddings (vs untied for LLaMa).
>  - Dataset: high-quality NVIDIA ClimbMix dataset (vs C4 used previously).
>
> We conducted training with identical model sizes, data-saturation ratios, and optimizer hyperparameters. We refitted all the scaling law parameters from scratch.
> We found that the new estimates for parameters (α,β,E,F,L,C) match the LLaMa/C4 fit: their point estimates coincide within the confidence intervals, obtained by the bootstrapping procedure described in our paper.
>
> |      | LLaMa/C4 | Transformer/ClimbMix10M (new) |
> |:---------|:--------:|:---------:|
> | $\alpha$  | $0.13 \pm 0.05$  | $0.18 \pm 0.03$  |
> | $\beta$  | $0.33 \pm 0.06$ | $0.26 \pm 0.02$  |
> | E  | $1.3 \pm 0.5$ |  $1.4 \pm 0.3$  |
> | L | $1.0 \pm 0.7$ |  $0.84 \pm 0.1$  |
> | F | $0.41 \pm 0.02$  | $0.37 \pm 0.04$  |
> | C  | $1.39 \pm 0.08$  | $1.24 \pm 0.13$  |
>
> This indicates that the same underlying law governs both model families and datasets. We will add those results in the revised version of the paper. We thank the reviewer for the suggestion.
>
>
> **4. The “effC” form of Frantar et al. [9]**
>
> We emphasize that the meaning of effC is subtly different between our work and that of Frantar et al. [9]: They treat all representation capacities as **independent variables** in the fitting process, and will naturally obtain lower error since it directly fits capacities for each bitwidth in turn.
>
> By contrast, assuming a functional form over them (such as our GMSE form or the inverse exponential of Kumar et al. [18]) effectively constrains the optimization problem, trading fit error for explainability and robustness of the law. The independent-variable “effC” solution provides the lower bound on fit error for the otherwise constrained solutions.
>
> From Table 1, one can see that the parametrization we propose achieves the error closest to the lower bound constituted by the “effC” form while also connecting capacity to GMSE. To further clarify this, we’ll expand the discussion around *Line 178*  with a discussion on how parametrizing capacity affects the scaling law fitting optimization problem.

---

> > ### Comment · Reviewer_ygts · 2025-08-01
> >
> > I thank the authors for clarifying the connections to Kumar et al and Frantar et al. The additional experimental results also strengthen the claims of the paper. I increase my score to Borderline Accept.

---

### Official Review · Reviewer_9G9f · 2025-07-05

**Clarity:** 2
**Significance:** 3
**Originality:** 4
**Rating:** 4
**Confidence:** 1

**Summary:**

The paper investigates the scaling law that predicts model performance when training occurs over compressed representations. It introduces a "representation capacity" metric that estimates parameter efficiency, based on fitting random Gaussian data. Experiments are provided to show that this capacity metric can predict model performance across different compression formats.

**Questions:**

N/A

**Ethical Concerns:**

["NO or VERY MINOR ethics concerns only"]

**Limitations:**

yes

**Paper Formatting Concerns:**

The paper title (in the PDF file) does not match what it shows on OpenReview.

**Quality:**

4

**Strengths And Weaknesses:**

Strengths:
- The paper provides a unified scaling law that incorporates various compression formats, including sparse, quantized, and hybrid representations.
- The proposed representation capacity metric is new.

Weaknesses:
- The paper is not easy to understand for readers not in the field of scaling laws and model compression.
- To my best understanding, there is no experiment showing how good the proposed scaling law fits the real data. I think a figure showing the predicted loss vs. the real loss would be helpful (similar to Figure 1 in [8] and Figure 7 in [18]).

---

> ### Author Rebuttal · Authors · 2025-07-30
>
> Thank you for your feedback!
>
>
> **1. Accessibility for readers outside scaling laws/model compression.**
>
> Thank you for pointing this out. We agree that the current presentation is quite technical, and, in the revision we will include a brief “reader’s guide” to our notation and assumptions. We expect these additions to make the paper more self‑contained.
>
>
> **2. Empirical fit of the proposed scaling law.**
>
> We fit the scaling laws identically to existing literature[1,2,3]: we minimize the Huber loss over the logarithm of the validation set loss with $\delta=1e-4$. The fits show MAE of around $6\cdot10^{-3}$ for validation loss prediction. We will include a visualisation of estimated vs actual loss in the revised version of the paper.
>
>
> **3. Title mismatch between PDF and OpenReview.**
>
> We apologize and will fix this discrepancy: the OpenReview title will be updated to match the PDF’s title.
>
>
> [1] https://arxiv.org/abs/2203.15556 [2] https://arxiv.org/abs/2411.04330 [3] https://arxiv.org/abs/2502.05003

---

> > ### Comment · Area_Chair_GYqo · 2025-08-07
> >
> > Dear Reviewer 9G9f,
> >
> > This is a gentle reminder that the extended Author–Reviewer Discussion period will conclude on **August 8 (AoE)**.
> >
> > At your earliest convenience, please read the authors' rebuttal, and actively participate in the discussion. Regardless of whether your original concerns have been fully addressed, we kindly ask that you:
> >
> > - Clearly state if your concerns have been resolved, or
> > - Specify what aspects remain unaddressed.
> >
> > Even if you believe no response is needed, please communicate this with the authors. **Staying silent is not acceptable.**
> >
> > Please note that the Mandatory Acknowledgement should only be submitted **after**:
> > 1. Reading the authors' rebuttal,
> > 2. Engaging in the discussion with the authors (and optionally other reviewers or the AC), and
> > 3. Completing your **Final Justification** and updating your score accordingly.
> >
> > Please do **not** click the "Mandatory Acknowledgement" button until all of the above are completed.
> > Reviewers who submit the acknowledgement without meaningful discussion will be flagged as non-participating under the Responsible Reviewing policy.
> > If I have a **strong** feeling that a fellow reviewer has not properly participated, I may flag the review as **insufficient**.
> > The program chairs will use these flags to determine whether a reviewer's own submissions may be desk rejected.
> >
> > Thank you for your thoughtful and timely contributions to the NeurIPS 2025 review process.
> >
> > Best regards,
> > Your AC,

---

> > ### Comment · Area_Chair_GYqo · 2025-08-07
> >
> > Dear Reviewer 9G9f,
> >
> > This is a gentle reminder that the extended Author–Reviewer Discussion period will conclude on **August 8 (AoE)**.
> >
> > At your earliest convenience, please read the authors' rebuttal, and actively participate in the discussion. Regardless of whether your original concerns have been fully addressed, we kindly ask that you:
> >
> > - Clearly state if your concerns have been resolved, or
> > - Specify what aspects remain unaddressed.
> >
> > Even if you believe no response is needed, please communicate this with the authors. **Staying silent is not acceptable.**
> >
> > Please note that the Mandatory Acknowledgement should only be submitted **after**:
> > 1. Reading the authors' rebuttal,
> > 2. Engaging in the discussion with the authors (and optionally other reviewers or the AC), and
> > 3. Completing your **Final Justification** and updating your score accordingly.
> >
> > Please do **not** click the "Mandatory Acknowledgement" button until all of the above are completed.
> > Reviewers who submit the acknowledgement without meaningful discussion will be flagged as non-participating under the Responsible Reviewing policy.
> > If I have a **strong** feeling that a fellow reviewer has not properly participated, I may flag the review as **insufficient**.
> > The program chairs will use these flags to determine whether a reviewer's own submissions may be desk rejected.
> >
> > Thank you for your thoughtful and timely contributions to the NeurIPS 2025 review process.
> >
> > Best regards,
> > Your AC,

---

### Decision · Program_Chairs · 2025-09-17

**Decision:**

Accept (poster)

**Comment:**

This paper investigates the scaling law that predicts model performance when training occurs over compressed representations.
A key contribution is the introduction of the "representation capacity" that quantifies the capacity of such representations to support learning, with the goal of predicting model performance across different compression schemes.
The authors provide empirical evidence to support the utility of this metric across several settings.

The reviewers are in agreement that the proposed idea of factorizing capacity scores is novel and thought-provoking.
The paper is generally well-structured and clearly written.
At the same time, reviewers noted that the empirical validation could be significantly strengthened, particularly through evaluation on a broader range of architectures and compression formats.

Overall, this work introduces an interesting perspective on measuring and understanding performance in compressed regimes.
I encourage the authors to extend their empirical study in the revised version to fully realize the potential of their framework.

I thus recommend acceptance.